# Physiological Changes as a Measure of Crustacean Welfare under Different Standardized Stunning Techniques: Cooling and Electroshock

**DOI:** 10.3390/ani8090158

**Published:** 2018-09-18

**Authors:** Kristin Weineck, Andrew J. Ray, Leo J. Fleckenstein, Meagan Medley, Nicole Dzubuk, Elena Piana, Robin L. Cooper

**Affiliations:** 1Department of Biology, University of Kentucky, Lexington, KY 40506-0225, USA; k.weineck@hotmail.de (K.W.); mcmedley92@gmail.com (M.M.); nicoleqdp@gmail.com (N.D.); 2Department of Medicine, Rostock University, 18055 Rostock, Germany; 3Division of Aquaculture, Kentucky State University, Land Grant Program, 103 Athletic Road, Frankfort, KY 40601, USA; andrew.ray@kysu.edu (A.J.R.); leo.fleckenstein@kysu.edu (L.J.F.); 4Biomedical Sciences, Eastern Kentucky University, Richmond, KY 40475, USA; 5Biochemistry, Western Kentucky University, Bowling Green, KY 42101, USA; 6Sea Farms Limited, Redditch, Worcestershire B98 0RE, UK; EPiana@sea-farms.com

**Keywords:** blue crab, crayfish, electric stunning, euthanasia, icing, shrimp

## Abstract

**Simple Summary:**

Physiological measures were examined during stunning of three commercially important crustacean species: crab, crayfish, and shrimp in an ice slurry or with electroshock. Neural circuits for sensory-central nervous system (CNS)-cardiac response and sensory-CNS-skeletal muscle were examined. Heart rate of shrimp was the most affected by both stunning methods, followed by crayfish, then crabs. Ice slurry and electroshocking may paralyze crabs, but neural circuits are still functional; however, in shrimp and crayfish the neural responses are absent utilizing the same protocols. The use of stunning methods should vary depending on species and slaughter method. Interpretation of behavioral signs should be supported by further research into related physiological processes to objectively validate its meaning.

**Abstract:**

Stunning of edible crustaceans to reduce sensory perception prior and during slaughter is an important topic in animal welfare. The purpose of this project was to determine how neural circuits were affected during stunning by examining the physiological function of neural circuits. The central nervous system circuit to a cardiac or skeletal muscle response was examined. Three commercially important crustacean species were utilized for stunning by immersion in an ice slurry below 4 °C and by electrocution; both practices are used in the seafood industry. The blue crab (*Callinectes sapidus*), the red swamp crayfish (*Procambarus clarkii*), and the whiteleg shrimp (*Litopenaeus vannamei*) responded differently to stunning by cold and electric shock. Immersion in ice slurry induced sedation within seconds in crayfish and shrimp but not crabs and cardiac function was reduced fastest in shrimp. However, crabs could retain a functional neural circuit over the same time when shrimp and crayfish were nonresponsive. An electroshock of 10 s paralyzed all three species and subsequently decreased heart rate within 1 min and then heart rate increased but resulted in irregularity over time. Further research is needed to study a state of responsiveness by these methods.

## 1. Introduction

In 2016, 6.7 million tons of crustaceans came from capture fishery and an additional 7.8 million tons were produced by aquaculture. While fishery capacity plateaued [1], output from aquaculture continue to rise [2] to meet the growing demand for seafood. Despite this tremendous market size for crustaceans in the food industry, there is not a standardized method for slaughtering crustaceans. Due to increased awareness in crustacean welfare, the application of electric stunning is currently receiving more attention for the humane slaughter for crustaceans [3]. Chilling in air and ice slurry are still the most common and practical ways of paralyzing and rendering crustaceans unresponsive. This study addresses the physiological impacts of chilling and electric stunning in key commercial species of crab, crayfish, and shrimp and focuses on bio-indices not commonly examined across species undergoing exposure to these stunning methods

The EFSA (European Food Safety Authority) concluded that decapod crustaceans like those subject of this study can experience pain and distress [3]. Evidence to support the claim that crustaceans can experience pain is mainly based on behavioral observations [4,5,6,7,8] and some more recent physiological measurements [9,10,11]. Policy-makers’ response has recently resulted in Switzerland and a province in Italy establishing new regulations on the ways of killing lobsters which resulted in banning some practices, amongst which, the one of boiling it alive. Nonetheless, some scientists still argue that it is difficult to provide evidence that crustaceans have this emotional capacity and awareness [12,13]. The conclusion that crustaceans can experience pain impacts both the scientific community and the seafood industry and this paper focuses on the latter. Several animal welfare organizations such as The Royal Society for the Prevention of Cruelty to Animals (RSPCA), the People for the Ethical Treatment of Animals, Animal Aid UK, Crustacean Compassion UK, and the Humane Society of the United States, advocate for increasing protection of crustaceans at the time of killing to minimize their perception of pain. Numerous approaches can be used to anesthetize and euthanize crustaceans such as freezing, superchilling (N_2_ gas), piercing of ganglia, salt baths (MgCl_2_), CO_2_, electric stunning, chilling in air or ice slurry, and boiling; each has advantages and disadvantages in terms of efficacy and animal welfare. Applicable methods which function at high efficacy with fast immobilization to reduce potential stress of crustaceans are sought after by the industry [14,15,16]. 

Electric stunning is well-established for stunning in finfish, and it seems to paralyze some crustaceans species (i.e., *Cancer pagurus*, *Homarus gammarus*, *Astacus astacus*, *and Astacus leptodactilus*) [10,11,17], but evidence of the effectiveness of electric stunning in other crustaceans is scarce and shrimp have not yet been a subject of specific studies. Some drawbacks of electric stunning are the induction of seizures in the central nervous system (CNS), the formation of blood clots in fish [18] and the spontaneous autotomy of limbs in crabs [8]. 

Although cooling may seem more practical in damping neuronal function in heterothermic invertebrates, past studies have indicated that decreasing the temperature only reduced the basal metabolic rate, with sensory information still being detected, processed, and integrated neurally in a rhythmic pattern [9,10]. In fact, some crustaceans (e.g. *Panulirus japonicus*, *Penaeus japonicus*, *Homarus americanus*, and *Ligia exotica*) have neurons which may detect cold since the neurons increase their firing rate in temperatures between 0.5 and 5.5 °C [10,19]. The regulation of the heart in the three crustaceans under study is neurogenic, meaning that each beat is controlled by neural signals from innervating neurons of the cardiac ganglion. The rhythm of the neural activity is controlled by a central pattern generator within the central nervous system of the animals. When a threatening sensory stimulus is introduced there is an alteration in the central neural regulation output to the cardiac tissue. The rhythmic control in higher brain centers of crustaceans which controls cardiac, respiratory, and digestive functions tend to maintain different frequencies at varying temperatures [20,21,22]. Cooling in ice slurry is recommended for tropical and temperate species that are susceptible to cold temperatures [23]. However, recommended immersion times are long (20 min) and some argue [24] that it is not yet demonstrated if the immersion induces paralysis but not anesthesia.

Stress, pain, and nociception are different states and it is complex to separate them from one another in invertebrates [13], as such their physiological measurement is difficult. Measurements within invertebrates such as behavioral avoidance, hormonal changes, and mobilization of energy stores such as glucose are not sufficient to define one state from another [12]. Nociception comprises the capacity of perceiving noxious stimuli via receptors and withdrawing from the stimulus through swift reflexes. “Pain” is an emotion that requires the capability of the animal to be aware of the noxious stimulus with the involvement of higher processing and consciousness [25]. Evidence provided so far on crustaceans is in some cases is contradictory. In the 1950s and 60s the researchers Baker [26] and Gunter [27] first described ways of slaughtering crayfish and crabs in a humane way and initiated a discussion about nociception in crustaceans. One of the observations made by Gunter was that slow heating to 40 °C in water for large crustaceans did not appear to cause the animals stress and would result in death. The same conclusion was also reached in the study by Fregin and Bickmeyer [10] but the EFSA [3] lists boiling live crustaceans as one of the methods that are likely to cause pain and distress. Barr et al. [4] demonstrated that irritating the antennae of prawn *Palaemon elegans* triggered a “tail-flick” response and they press their antennae against the walls of the enclosure which could be a behavioral means to remove the irritant from the antenna. However, Puri and Faulkes [28] found no behavioral or electrophysiological evidence that antennae contained nociceptors for extreme pH or benzocaine/ethanol. Carbon dioxide can be used to anesthetize crustaceans before euthanasia, but it is now known that crayfish will avoid water tainted with CO_2_ and that CO_2_ reduces pH in the water [29]. Since crustaceans and insects appear to have a functionally similar autonomic nervous system as mammals [30,31,32,33,34] it is not surprising that responses to an aversive stimuli would be analogous to those of vertebrates in moving away or retracing from the stimuli [5,6,34,35].

Studies indicate that different stunning techniques cause different physiological responses to stimuli. Lactate levels increase almost three-fold in crabs that are electroshocked [36]. Other compounds such as biogenic amines and neurotransmitters (i.e., serotonin, dopamine, and octopamine) rise in crabs with exercise causing alterations in behavior [37,38,39]. However, one needs to be cautious in relating changes in the compounds in the hemolymph to one condition such as exercise, environment, or stunning procedure, as these compounds can vary with a molt cycle, circadian cycle, gravid status, social dominance, and health of the animals [15,16,40,41,42]. Even insects show changes in levels of biogenic amines with environmental stressors and/or exercise [43,44,45]. In cold (10 °C) exposed *Drosophila melanogaster* both serotonin and octopamine concentrations decreased in the hemolymph [46]. Crayfish gradually exposed to cold (20–21 °C to 15 °C for 1 week and one week at 10 °C) increased hemolymph concentration of octopamine 4-fold [46]. No studies that we are aware of have addressed if a rapid exposure to cold would also raise octopamine or alter levels of serotonin. Since it is known that the levels of these compounds can increase quickly with exercise, it was feasible to expect some changes could occur quickly with rapid cold exposure. With electroshock we predicted an increase in the level of the serotonin and octopamine due to electrical stimulating the neurons to release these substances into the hemolymph. In contrast, with rapid exposure to cold we did not expect any change in the levels in the hemolymph due to decreased activity of the neurons to release these substances. In this experimental study design, we measured the concentration of these two commonly assayed compounds (serotonin and octopamine) among three crustacean species using the same measurements techniques with environmental changes to provide a baseline for other investigators and comparative studies.

The aim of this study was to determine the effectiveness of electroshock and thermal shock by ice slurry for the stunning the blue crab (*Callinectes sapidus*), the red swamp crayfish (*Procambarus clarkii*), and whiteleg shrimp (*Litopenaeus vannamei*) through measurements of physiological responses. Physiological measures consisted of changes in heart rate (ECG), neural activity to external stimuli by alteration in the heart rate, and changes in the levels of biogenic amines within the hemolymph. The effects on skeletal muscle activity by electromyograms (EMG) of the large closer muscle in the chela were measured in crabs and crayfish. In finfish, the effectiveness of different stunning techniques could be demonstrated by the measurement of the electroencephalogram (EEG). However, due to the impractical nature of recording an EEG in crustaceans, as reliably performed in large fish, heart rate may be recorded as a reliable bio-index. The hearts of shrimp, crab, and crayfish are neurogenic, meaning that the rate of beating is indicative of the neuronal function [34,47]. The words ‘paralysis’ is used to indicate absence movements but not absence of ECG and EMG measurements. The word ‘anesthesia’ indicates absence of neural function, thus absence of perception of sensory stimuli. 

## 2. Materials and Methods 

### 2.1. Animals 

Experiments were performed using red swamp crayfish (*Procambarus clarkii*, Atchafalaya Biological Supply, Raceland, LA, USA), blue crab (*Callinectes sapidus*, food distribution center in Atlanta, GA, delivered to and bought from a local supermarket in Lexington, KY, USA), and whiteleg shrimp (*Litopenaeus vannamei*, Kentucky State University Aquaculture Research Center, Frankfort, KY, USA as well as from Belize Aquaculture Ltd., Mile 4 Placencia Road, Stann Creek District, Belize).

Throughout the study, midsized crayfish measuring 6–10 cm in postorbital carapace length (posterior dorsal surface of the orbital cup to the end of the carapace directly posterior to the eye cup) were used. The measures were made with calipers (Swiss Precision, Newton, MA, USA, 0.1 mm). The animals varied in weight from 12.5–25 g. They were individually housed in standardized plastic aquaria (33 cm × 28 cm × 23 cm, water depth 10–15 cm) with temperature maintained between 20 and 21 °C, weekly water exchanges, constant aeration, and dry fish food provided every 3 days (salinity 25–26 ppt; O_2_ at 7.4–7.6 mg/L). To ensure the vigor of the blue crab, they were held in a seawater aquarium prior to use for 3 to 5 days. All experiments were implemented in female adult crabs with a carapace width (from point to point) of 10–15 cm and a body weight of 140–225 g. The crabs were fed with frozen squid every 3 days (cannibalism also occurred) and the water temperature was maintained between 20 and 21 °C (salinity 0.2–0.5 ppt; O_2_ 7.5–7.9 mg/L). Shrimp were raised at the Kentucky State University Aquaculture Research Center (KSU) for 85 days in aerated water between 27 and 28 °C and fed with commercial shrimp feed (salinity 15 ppt; O_2_ 7.35–7.7 mg/L). The aquaculture system used was a modified biofloc system that allowed the growth of bacteria in the system to control water quality and detoxify waste products [48,49]. Studies in Belize used shrimp raised in outdoor open ponds in a large-scale aquaculture farm. They were transferred to an open window laboratory ranging in water temperatures from 30 to 31 °C. Shrimp with a postorbital carapace length of 20 to 35 mm and body weight 23.5 g were used from KSU and 25–38 mm from Belize.

Care was taken not to use more animals than necessary for these studies. According to University of Kentucky Administrative Regulation (AR) 7:5, oversight applies to “all research, teaching, and testing activities involving vertebrate animals conducted at University facilities or under University sponsorship, regardless of the species or source of funding.” This AR follows The United States Department of Agriculture (USDA) Animal Welfare Act, the PHS Policy and the *Guide* definition of animal. With this in mind, The Institutional Animal Care and Use Committee (IACUC) review is not currently required in the US for the use of crayfish, crabs and shrimp in research.

### 2.2. Electromyograms (EMG) and Electrocardiocrams (ECG)

The preparation of the ECG and EMG leads is described in detail in text and video format in previous publications [50,51]. In brief, insulated stainless steel wires (0.13 mm diameter; A-M Systems, Carlsburg, WA, USA) were inserted into the small holes made in the cuticle (Figure 1). For heart rate measures in all three species the wires were placed through the dorsal carapace directly over the heart [52]. To eliminate the risk of damaging internal organs, special attention was made on inserting only a short portion of wire (Figure 1). 

For ECG recordings, both wires were connected to an impedance detector (UFI, model 2991, 545 Main Street, Suite C-2, Morro Bay, CA, USA) which measures dynamic resistance between the leads. Subsequently, the detector was linked to a PowerLab/4SP interface (AD Instruments, Unit 13, 22 Lexington Drive, Bella Vista, New South Wales, Australia) and calibrated with the PowerLab Chart software version 5.5.6 (AD Instruments). The acquisition rate was set to 10 kHz. The calculation of the heart rate was accomplished by direct counts of each beat over short 10–20 s intervals and converted into beats per minute (BPM). The responsiveness of a sensory-CNS-cardiac ganglion neural circuit was assessed using a wooden rod to tap the dorsal carapace of the crab and crayfish inducing an alteration in the heart rate [32,51,52]. A physical pinch by an experimenter using forefinger and thumb was periodically made on the telson of the shrimp to induce an ECG response.

The EMG myographic recordings were performed in crayfish and crabs using two stainless steel wires placed in the closer muscle in the chela (Figure 1). A third wire located in the carpopodite region of the same limb served as a ground lead [54]. Similar to the ECG recording procedure, the holes in the cuticle were formed and wires prepared, inserted, and fixed in the respective area spanning the closer muscle in its central region. Via a Grass AC preamplifier (P15; Grass Instruments, Astro-Med Industrial Park 600 East Greenwich Avenue, West Warwick, RI, USA) the potentials were detected differentially and acquired digitally as previously described [54]. To elicit high frequency responses in the EMG signal, the crabs and crayfish were teased using a wooden rod placed in the jaws of the chela. Rubbing on the teeth of the chela produces the reflexive action to the motor neurons to produce the gripping response. Thus, a sensory-CNS-motor circuit is recruited by the rubbing action on the teeth of the chela [55,56].

### 2.3. Ice Slurry

In order to test the physiological changes of cooling crustaceans, we placed the animals for 5 min in plastic boxes containing crushed sea water ice (shrimp and crabs) or freshwater ice (crayfish) which was the same water the animals were maintained in prior to experimentation. The temperature in the boxes was between 0 and 4 °C. 

### 2.4. Electrical Stunning

Electrical stunning was performed using an AC source (60 Hz, 120 Volts, 20 amps) with wires directly from the wall outlet attached to two carbon rods (12 cm length × 1.3 cm dia.). The animals were transferred individually into a plastic chamber (17 cm × 12.5 cm) with the two carbon rods along each long side of the container submerged half the depth of the water level. This is so the rods were not resting on the bottom. Crayfish were shocked in a 1:1 mixture of freshwater:seawater (same type of waters described above). Initial trials with shocking crayfish in fresh water did not paralyze the animals. The 1:1 mixture enhanced electrical conductivity and resulted in paralysis within the 10 s window. Crabs and shrimp were shocked in seawater. Animals were electrocuted for 10 s and observed for behavioral changes. The animals were rapidly moved to their previous container for further measures of signals in the ECG and EMG traces. These animals were not previously exposed to any other experimental treatments besides electric stunning.

### 2.5. Hemolymph High Pressure Liquid Chromatography (HPLC) Samples

To evaluate if changes in hormonal levels might occur with crustaceans rapidly exposed to an ice slurry and electroshocking, approximately 0.5 mL hemolymph was drawn from six crayfish and six crabs and between 0.2 and 0.5 mL of hemolymph from six shrimp. The hemolymph was obtained directly in the hemocoel with an 18 gauge needle either in a ventral puncture close to the ventral nerve cord (shrimp and crayfish) or in the basal joint of the last walking leg of the crab. The only hemolymph samples from shrimp were from those at KSU. Control hemolymph samples were taken from animals exposed to their respective environment and temperature without being exposed to ice slurry and electrostunning. The hemolymph was mixed 1:1 in the tube containing the HPLC mobile phase and immediately frozen and stored at −80 °C until HPLC could be performed. The quantification of serotonin (5-HT) and octopamine levels in the hemolymph was accomplished through high pressure liquid chromatography with electrochemical detection (HPLC-EC). The samples were analyzed at the Center for Microelectrode Technology (CenMeT) and Parkinson’s Disease Translational Center of Excellence, University of Kentucky Medical Center, Lexington, KY, USA. 

### 2.6. Statistical Analysis

All data are expressed as mean ± SEM. The rank sum pairwise test or a sign test was used to compare the difference of heart rate with exposure to ice slurry or electroshocking. The nonparametric tests were used because data were not normally distributed as there was no activity to measure in some conditions when heart rate stopped. ANOVA and posthoc analysis were also conducted on some data sets. This analysis was performed with SigmaStat software. A *p*-value of ≤ 0.05 was considered statistically significant.

## 3. Results 

### 3.1. Habituation Rate in Shrimp Tail Flipping to a Stimulus

We induced threatening stimuli by physical pinches to the telson on the tail of the shrimp before and during cold exposure. Since such sensory stimuli can habituate over repetitive trials in crayfish [38], we set out to test the habituation time in shrimp by pinching the telson every 30 s until they stopped tail flipping. Habituation was defined as the time it took the animal not to tail flip in three repetitive pinches. On average it took 16 stimuli before habituation was present (Figure 2). Since the animals were only stimulated once or twice before cooling and once or twice during cooling the animals were not likely habituated to the sensory stimuli while examining for changes of the heart rate in the different temperatures. 

### 3.2. Effect of Cold Shock on Heart Rate and Response to a Sensory Stimulus

A representative ECG trace of a shrimp from Belize shows that within 10 s the heart rate has dropped, and the amplitude of the signal has decreased (Figure 3A). Upon transfer back to warm water (30.5 °C) the heart rate increases. The enlarged views of the ECG trace upon cooling (Figure 3B), in ice slurry (Figure 3C), and warming (Figure 3D) are highlighted. Within 15 to 30 s the heart rate is not measurable (Figure 3C). The amplitude of the signal is not the direct electrical measure of the heart as in standard ECG measures as one might be used to seeing for mammals. This ECG trace is an impedance measure which is basically measuring any movement of tissue (i.e., the heart) or fluid between the two wires causing a disturbance in the minute electrical field which is induced between the two leads by the amplifier. This measure has proven to be more sensitive than field recordings of electrical activity of these small hearts in crustaceans and will detect movements in the disturbance of the fluid between the wires [52,57]. Thus, the decreased amplitude relates to a reduced strength of contraction and likely, at the same time, reduced hemolymph flow. The rate in occurrence of beats is directly related to the rate of contractions. After 90 s for this particular shrimp the signal is completely undetectable (Figure 3C).

The effect on heart rate during immersion in cold water resulted in a reduced heart rate in all three species (each species; *N* = 6, *p* = 0.03 nonparametric sign test, two-tailed). The three species showed variability in the extent and rate of the decreasing heart rate with cold exposure. Even after 5 min of being immersed in an ice slurry, the majority of the crabs still had a pronounced heart rate, although the rate decreased in all six crabs (Figure 4A). Crayfish decreased to a similar rate to shrimp in about 1 min (Figure 4B). Despite a decrease in heart rate in shrimp they would tail flip within 1 min when exposed to the cold water. The heart rate of both species recovered after being placed back in warm water. The time of exposure in warm water was monitored longer for crabs to observe if a steady rate would be reached. However, both crabs and crayfish showed a rapidly increasing rate within 2 min. None of the crabs or crayfish died from the acute cold exposure. 

The shrimp from Belize had a higher average heart rate than the shrimp from Frankfort, KY but there was a substantial range in the values of heart rate among both sets of shrimp. The shrimp from Frankfort, KY were all maintained for 4 min in the cold before being warmed. Also all eight shrimp from Frankfort, KY significantly decreased their heart rate with exposure to cold (Figure 4C, *N* = 8, *p* = 0.008 nonparametric sign test, two-tailed). The water temperature was higher in Belize (~30.5 °C) compared to Frankfort, KY (27–28 °C) likely contributing to the average higher average basal heart rate in the shrimp from Belize. The time of maintaining the shrimp from Belize in the ice slurry varied before they were placed back in the warm water. When the heart rate was essentially stopped the animals were removed from the cold and placed in warm water to determine if the cold shock had killed them since the heart rate ceased. All six of the shrimp from Belize decreased significantly their heart rate with cold (*N* = 6, *p* = 0.03; nonparametric sign test, two-tailed). Within 30 s the decreased in the heart rate is substantial (Figure 4(D1)). The rate of rise in heart rate after being warmed was very quick for the shrimp in Belize even for ones held for at least 4 min in the cold (Figure 4(D2)). Although the heart rate did increase for most of the shrimp used at Frankfort, KY the rates were slow and in one case it appeared an animal had died from this handling procedure or exposure to cold. The effect of cold immersion did significantly reduced heart rate in shrimp from both locations. 

### 3.3. Heart Rate Changes Triggered by Sensory Stimuli before and after Cold Shock 

The effect of the sensory stimulus resulting in a pause in heart rate, prior to cold exposure, was present for all three species (*N* = 6, *p* = 0.03 nonparametric sign test, two-tailed). Crabs being immersed for 2 min still showed a marked response to a sensory stimulus (Figure 5(A2)); *N* = 6, *p* = 0.03 nonparametric sign test, two-tailed). Crayfish were comparable to the shrimp in the rate of decreasing responsiveness to sensory stimuli in the cold, both did not give as a pronounced response as crabs (Figure 5, A2-Crab, B2-Crayfish, C2-Shrimp). Shrimp showed a pause in heart rate with a sensory stimulus which was likely perceived as threatening (Figure 5(C1)). We are not aware of this response being reported previously for shrimp, but this phenomenon has been reported to occur for crayfish of the same species and a crab of a different species [32,51,52,58]. However, after just 30 s in the cold no detectable pauses or consistent alterations in the heart rate for the crayfish or shrimp could be detected when pinched or tapped (Figure 5(B2,C2)). This was shown for six preparations in each of the three species (*N* = 6, *p* = 0.03 nonparametric sign test, two-tailed). Upon warming for 3 to 5 min, after a cold shock, when pinched, the pauses in the shrimp were not as prominent as prior to the cold shock; whereas, in the crab they were still obvious (Figure 5(A3)). It should be noted that the sensory stimuli were not given at consistent intervals upon warming as the rates of showing an increase heart rate varied. After rates increased in amplitude well enough to be observable this is when a tap on the carapace (crab and crayfish) or a pinch to the telson (shrimp) occurred.

### 3.4. Heart Rate Measures before and after Electric Stunning

The electric stunning produced a variety of responses. Crabs would just remain immobile. The crayfish flexed their abdomen and remained in a flexed position for duration of the shock as did most of the shrimp. Shrimp in some cases would tail flip so violently they could flip themselves outside the shocking tank. In these cases, the heart rate data was not used as they were not shocked for a consistent 10 s. The amplifier is turned off during the electric shock to protect the equipment. The electric shock lasts for 10 s and the amplifier is turned back on to record the ECG right after the shock. As shown for all three species, heart rate became arrhythmic immediately after the electric shock was terminated Figure 6(B2,C2,D2). The shapes of the ECG traces did not regain full recovery to the state they were prior to the shocking. This is an indication of some permanent damage or alteration in the function of the heart Figure 6(A3,B3,C3); *N* = 6, *p* = 0.03 nonparametric sign test, two-tailed).

The heart rates prior to electric shock and just afterwards revealed that 10 s of shocking decreased heart rate in all cases for the three species (Figure 7). As with cold shock, the heart rates dropped most substantially for shrimp (Figure 7C). In two crabs the shocking killed the animals as the heart rate did not return and the animals remained unresponsive to stimuli and limp. Due to the instability in the ECG traces for some of the crayfish, heart rates were not determined right away after shock but were available after 1 min passed. Heart rate rose after the electric shock was over for all of the living animals. Even though the measures were not immediately available for all crayfish, all animals were immobile as with all crabs and shrimp followings the shock. It took 5 to 10 min before the animals gained some ability to move or swim. Only the shrimp from Frankfort, KY are represented in the electric stunning data. Preliminary studies were first conducted with shrimp in Belize but a large enough sample size was not available for quantitative purposes. In these preliminary studies it was found that electrocution longer than 10 s caused heating of the animal and surrounding water, which lead us to adopt 10 s electrical intervals in the studies performed in Kentucky. Likewise, electric shocking of the crabs for more than 10 s produced heating of the animal [17]. In all cases there were decreases in heart rate observed with 10 s of shocking for all three species (*p* < 0.05 nonparametric sign test). 

### 3.5. EMG Measures before, during, and after Cold Shock

To further assess the effects of cold on physiological functions, a neural sensory-CNS-motor circuit was examined in crab and crayfish but not shrimp due to the smaller chela. The sensory stimulation of the teeth on the chela produces robust activity in the muscles of the closer muscle as shown in the EMG traces for crab (Figure 8(A1)) and crayfish (Figure 8(B1)). During cold exposure the responses are greatly attenuated in both species (Figure 8(A2,B2)). Upon warming the animals, after a cold immersion, the EMG activity would return (Figure 8(A3,B3)). This was present in all of the crabs and crayfish examined (*N* = 6 each species, *p* = 0.03 nonparametric sign test, two-tailed). 

### 3.6. Modulators in the Hemolymph during Cold Shock and Electric Stunning

Analysis of commonly identifiable neuromodulators (octopamine and serotonin), which serve as indicators of humoral changes for physical activity, were examined from hemolymph samples obtained in control animals not exposed to cold or to electric stunning and in animals exposed to cold for 5 min or to electric stunning for 10 s. The HPLC analysis showed a greater concentration as well as a larger variation for crab in both octopamine and serotonin compared to crayfish and shrimp (Figure 9). Both crayfish and shrimp had low levels of serotonin and octopamine compared to crab. There were no detectable differences in serotonin and octopamine for crab or shrimp with cold shock or electric stunning. The hemolymph of crayfish could not be analyzed fully as the levels of octopamine were below the levels of HPLC detection. Since octopamine could be measured at room temperature this may indicate a reduction in their levels due to cold and electric shock. Statistical analysis on detectable levels for the six crayfish as compared to not being able to measure the levels with cold shock and electric stunning produces a non-normal distribution but significant difference by Kruskal–Wallis test (*p* = 0.01). Serotonin levels were lower for cold shock (*p* = 0.028, *N* = 6, Holm–Sidak) and electric stunning (*p* = 0.004, *N* = 6, Holm–Sidak) in crayfish as compared to 20 °C but not for crabs and shrimp.

### 3.7. Isolated Shrimp Abdomens Show Tail Flipping

To examine if the tail flipping in shrimp exposed to the ice slurry is induced by the central ganglia six shrimp in Belize were rapidly transected at the abdomen and cephalothorax and both halves were placed in the ice slurry. To maintain animal welfare crustaceans are not normally decapitated without prior anesthesia, but in this case it was necessary to have a fully awake animal to test the idea; the decapitated head of the animal was immediately placed in the ice slurry. In all six animals, the abdomen was not flipping upon being placed in the ice slurry but started tail flipping in 1 min. Thus, the tail flipping is reflexive within the abdomen and cognitive function is not involved. When the segmental roots are cut along the ventral nerve cord and the abdomen placed in an ice slurry the tail flipping does not occur. Thus, the response is neural within the ventral nerve cord and not the muscle itself or a burst of activity of the axons or nerve terminals of the phasic-type of motor neurons innervating the flexor muscles in the abdomen. This response indicates that the flipping and contractions are involuntary movements, not driven by coordinated activity in higher central brain centers but within the ventral nerve cord within the abdomen. The neural circuit of the tail flip is well-established and it is known that the motor nerve output to stimulate the muscles can be locally recruited within the isolated abdomen [59,60,61,62,63,64]. In the Appendix A videos of whole shrimp are shown being placed in an ice slurry and after 1 min they start tail flipping (S1). When the abdomen is removed from the animal and placed in the ice slurry the same phenomena occurs (S2). However, when the motor roots are cut from the ventral nerve cord to the muscles no tail flipping of the isolated abdomen occurs (S3). 

## 4. Discussion

This study highlighted the physiological effects in three commercially important species of crustaceans for cold ice slurry immersion and electric stunning as a potential means of the stunning in the seafood industry. Crabs showed the least response to chilling, demonstrated by maintaining a sensory-CNS-cardiac or sensory-CNS-skeletal muscle response with cold for 4 min, and did not decrease their heart rate as quickly or as much with chilling as did shrimp or crayfish. Crayfish followed in the rate of decreased HR in the same manner for the physiological responses as shrimp during the rapid cooling. Shrimp showed the fastest and most pronounced decrease in heart rate, as well as a sharp decrease of activity within a sensory-CNS-cardiac response when chilled, and a quick rise in heart rate upon warming. The decrease in neural activity from the cardiac ganglion driving the heart occurred in shrimp as soon as 5 s in some cases (Figure 3B). Electric stunning for 10 s with 120 V AC did not produce burning marks on the crustaceans and like cooling, it did not kill the animals. All three-species appeared stunned as they did not show coordinated responses to stimuli after electroshock. Although results are not presented, shrimps were stimulated by touching the eye stalk and no response was evident. The bioindex of monitoring heart function indicated the heart was still beating although compromised after electric stunning. This finding is consistent with the results on electro stunning presented by Fregin and Bickmeyer [10]. 

The EMG activity measures in muscles can detect the electrical activity related to muscle contraction; however, if the muscle is induced to contract only a low frequency of basal activity will be detected unless the animal induces a response. In this study, the EMG activity was induced by stimulating a sensory response by rubbing the teeth on the chela of crabs and crayfish [55,56]. The sensory stimulation is integrated in the CNS to then activate motor neurons and appears to be reflexive in nature. The decrease in the activity with cold indicates this neural reflex is dampened and the sensory responses may not even be detected. Likewise, according to these results the sensory-CNS-cardiac system is assumed to be reflexive and is greatly dampened with cold, particularly for shrimp as compared to crabs. Thus, high neuronal function in processing the sensory input within the central brain (i.e., cerebral ganglion) and subesophageal ganglion and the production of a functional motor output is lacking. This type of autonomic reflex in altering heart rate is common from crustacean to higher mammals such as primates. Even though the heart in mammals is myogenic there is robust neuronal control, however a similar response is observed with a slight pause in the heart beat with an acute threatening sensory stimulus [30,32,53,57,65,66,67]. Because this was reduced in shrimp with immersion in an ice slurry it may indicate that this approach reduces a perceived stimulus which would activate an autonomic response. This indicates that an ice slurry may be an effective method for stunning crayfish and shrimp without increasing nociception. 

The ecology of the three species may help explain the reactions to ice slurry. Crayfish are known to maintain the sensory-cardiac response at 10 °C and even down to 5 °C for over a week when slowly acclimated to cold for 2 weeks [9]. *Procambarus clarkii* is native to the southern United States but is an invasive species now found in different regions from the Great Lakes to the Scandinavian Peninsula and Japan [68]. This wide distribution in the wild may have come about from the ability of this species to acclimatize to warm and cold environmental conditions. Likewise, the blue crab range is from Nova Scotia to Argentina and exists in Asia and Europe [69]. This crab is considered eurythermal and may well be able to resist decreasing neural function upon acute cold exposure due to cold acclimatization. More thorough studies of *Procambarus clarkii* and *Callinectes sapidus* with physiologic measures would help to uncover their abilities to function in acute and chronic cold conditions. In contrast, the shrimp *Litopenaeus vannamei* has a tropical and subtropical distribution, and it is not recognized as an invasive species despite being widely introduced to non-native locations for commercial purposes [68]. This may be partly due to its inability to adapt to low temperatures. The tropical prawn species *Macrobrachium rosenbergii* is also very sensitive to cold and loses the sensory-cardiac response with acute cold. This prawn species will die with acute or slowly induced cold to 5 °C [9].

Electric stunning is harder to induce with consistency than cooling in an ice slurry due to the nature of electricity. The resistance of the water being fresh or seawater conducts the current differently around an animal in a holding tank. In addition, the salt concentration of an animal’s hemolymph, size of animal, and dimensions in respect to the electrical conducting leads will result in different amounts of current each animal will be exposed to. Since the resistance or impedance of the animals vary, there are several parameters to be considered with electrostunning. It is complex to deliver a consistent and appropriate level for preserving quality of the muscle and inducing paralyzes with different species and sizes of crustaceans [8]. If an animal is suspended in the water or in air, placing the contacting leads needs consideration to provide consistent electrical conduction through the animal. We provided enough alternating current at 120 V to make the animals motionless and to be visually assumed dead. However, with shrimp, upon close observation through the almost translucent cuticle, it was easy to observe the scaphognathite within the prebranchial chamber to monitor the ventilatory rate [51,70]. This ventilatory organ continued to beat after the electric stunning and one can see the heart beat with good lighting from the side of the animal. Thus, even though the animal appears physically paralyzed with electric stunning, physiological functions like breathing and heart beating are still active. Further analysis is needed to determine the ability to process external stimuli by measuring changes in heart rate and how long and intense an electroshock is needed for inducing lack of responsiveness of the sensory-CNS-cardiac ganglion. We postulate that electroshock induces effective stunning as the animals take about 5 to 10 min to regain some ability to move or swim after application of the current. This would provide some time for a processing facility to handle the animals before slaughter or placing them on ice to die, but without a standardized length of time for unresponsiveness, the handling would have to be closely monitored to ensure welfare. 

One may visually monitor the beating of the scaphognathite more readily in crayfish and crab if part of the lateral cuticle would be removed [70]. The beat rate can be monitored with impedance measures as performed to record heart rate in this study, but the signals vary greatly depending on the placement of the recoding leads [50,51]. The observations that electrical shocking induced paralysis was similarly reported by Fregin and Bickmeyer [10] in studies with crayfish and lobsters. Edible crabs of Norway were shocked for 10 to 20 s at 220 V and still many of them were still moving [8]. More thorough studies in the timing and a means of delivering a consistent electrocution for different sizes and species of crustaceans without compromising the muscle by heating or scarring are needed [8]. Wild caught animals will likely present a wider size range than farmed crustaceans for food processing and crabs and crayfish may even be missing some of their limbs which could vary the capacitance of the animal which affects the electric stunning. 

The concentration of the biogenic amines serotonin and octopamine are higher in crabs than crayfish and shrimp. The decrease in octopamine in the crayfish with rapid cold exposure was unexpected since an earlier study showed an increase in octopamine in crayfish during a prolonged cold exposure of 2 weeks [46]. Perhaps a longer time is needed for the levels to demonstrate changes. Serotonin levels decreases for cooling and electric stunning in crayfish but not statistically significant in the other species. The biological reason this decrease is not clear and the mechanism driving the decrease in the crayfish has not been established. Further investigations are needed for establishing longer term monitoring of biogenic amine levels at various times after cold shock and electric stunning. The hemolymph measures of serotonin and octopamine are not very telling for the paradigms used in this study except that crabs apparently have higher concentration of both compounds in their hemolymph and that there appears to be a lot of variability among animals under the same conditions. The low levels in hemolymph of crayfish and shrimp made it difficult for comparisons. An approach in the future which may be beneficial would be to obtain as much hemolymph as possible and lyophilize it and resuspend in a smaller volume to concentrate the compounds or to make use of mass spectrometry-based approaches to screen peptides as well [71]. Even the handling of the animals to draw hemolymph samples may result in the animals altering the levels of the compound to be measured [14,72,73,74,75]. Both serotonin and octopamine can increase central neural activity in crustaceans and insects and in some cases synaptic transmission at neuromuscular junctions. Such biochemical measures in crustaceans are scant for effects of potential stressors of different types such as electric stunning or cold shock. 

### Animal Welfare Implications

These studies indicate that depending on the species of crustacean the effects of different stunning procedures will vary, and also that visual observations on their own are in some cases unreliable to assess stunning effectiveness. A few seconds of exposure to electric stunning does paralyze shrimp, crab, and crayfish which may improve the welfare during slaughter as the next step. It is likely that electroshock renders these crustaceans unable to perceive and elaborate stimuli for a period of time after shocking is applied but we did not test the perception of tapping/pinching stimuli on the animals to measure heart rate changes during or right after electric stunning due to human safety issues with the capacitance charge still present in the tanks and further investigation is needed.

With exposure to ice slurry, the responses vary. Anesthesia takes longer for crabs as noted in the responsiveness in pauses in heart rate to external stimuli. Most of the crayfish with acute cold exposure are also insensitive to sensory stimuli after 2 min when their heart rates have decreased substantially but crabs maintained central neural processing for cardiac and skeletal muscle reflexes. Crayfish and shrimp responsiveness to sensory stimuli stopped after just a few seconds. As shown in the supporting videos, suggesting a means of establishing unresponsiveness by verifying absence of resistance to handling and ease in manipulating body parts, such as the tail, does not seem to be validated, as tail flipping of isolated abdomens produces a coordinated movement even though in the intact animal the heart has stopped and is nonresponsive to sensory stimuli while tail flipping continues to occur. 

Future research should determine how long shrimp should be maintained in ice slurry so they do not recover when they are warmed up. It may not appear that shrimp lose sensory-CNS-skeletal muscle activity when put in an ice slurry since exposing them to an ice slurry induces them to tail flip and contract violently in about 1 to 2 min. However, it appears cooling in ice is an effective anesthetic for sensory perception. 

As legislation on the protection of animals at the time of death becomes more complex and more attention is drawn to animal welfare of all species, including crustaceans, more research to ascertain what the physiological responses are to different stunning methods will be required to support behavioral studies. The Scientific Panel on Animal Health and Welfare (AHAW) [3] endorses chilling in an ice water slurry, chilling in air, immersion in clove oil bath, and electroshock as humane means to kill decapods with minimal distress. However, this panel also raised the point that pain and distress in crustaceans may occur by placing animals in cold water and heating the water to boiling point. More recent studies of crayfish by Puri and Faulkes [76] confirmed the presence of nociceptors responsive to heat, but did not find evidence of nociceptive responses to low temperatures. The same panel and the RSPCA [23] also states that all crustaceans should be immersed in ice slurry for at least 20 min to reach unconsciousness whereas here we demonstrate that for crayfish and shrimps a few seconds are enough to anesthetize them. This is why recommendations should be based on rigorous scientific inquiries, rather than solely on behavioral observations or assumptions [12,77,78]. In the future, policy makers may benefit in adopting some of the presented methods for assessing perception of obnoxious stimuli in crustaceans to ensure animal welfare, giving consumers confidence that they are purchasing ethically produced seafood products.

## 5. Conclusions

Stunning of edible crustaceans to reduce sensory perception prior and during slaughter is a consideration in animal welfare. The blue crab (*Callinectes sapidus*), the red swamp crayfish (*Procambarus clarkii*), and the whiteleg shrimp (*Litopenaeus vannamei*), responded differently to stunning by cold and electric shock. The immersion in ice slurry induced sedation within seconds in crayfish and shrimp but not crabs. The neural drive on cardiac responses was reduced the quickest in shrimp. Crabs maintained a neural circuit over the same time when shrimp and crayfish were nonresponsive. The 10 s electroshock provided in these studies paralyzed all three species and subsequently decreased heart rate within 1 min. More research is required to address the varied sizes of animals, conditions the animals are raised in, and methods of delivering electric stunning. Regulations of crustacean stunning needs to be based on scientific measures of physiological functions and not only on behavioral observations.

## Figures and Tables

**Figure 1 animals-08-00158-f001:**
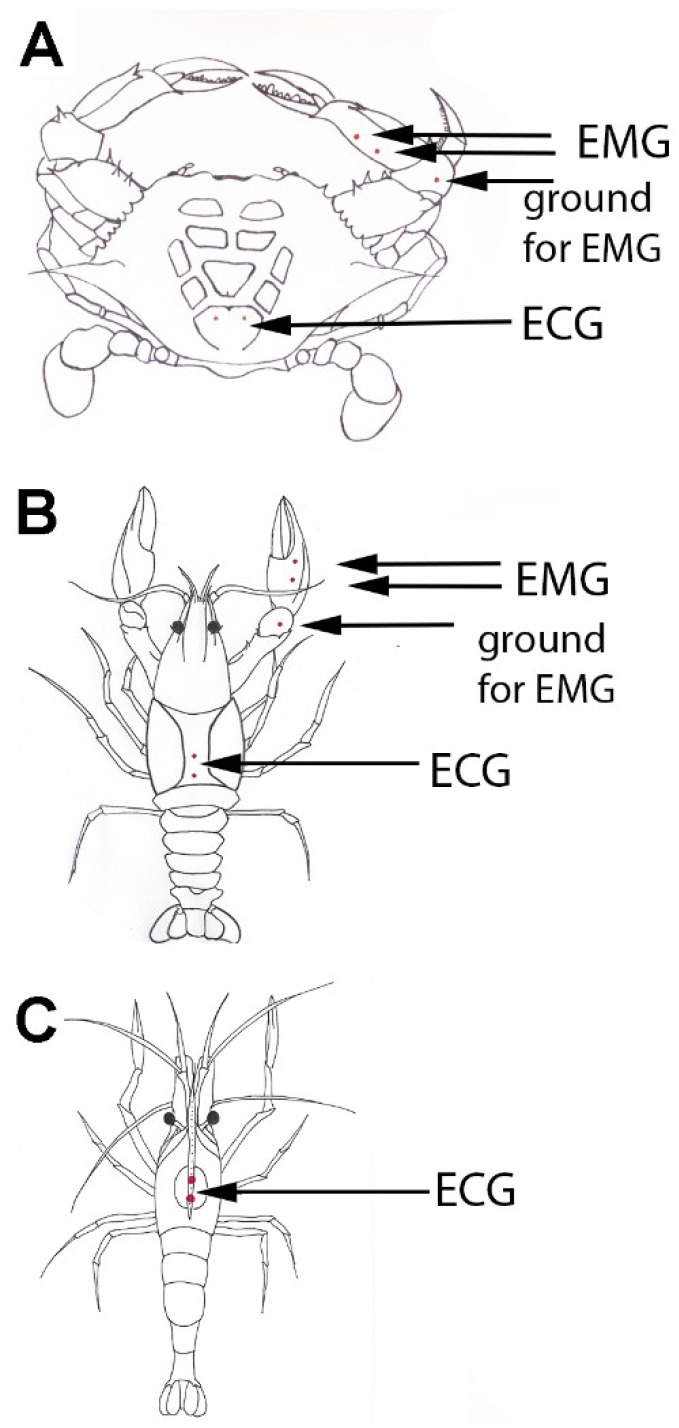
Placement of recording leads for measuring heart rate in an electrocardiogram (ECG) and skeletal muscle activity in an electromyogram (EMG) for the crab (**A**), crayfish (**B**) and shrimp (**C**). The two differential EMG leads to record the EMG activity of the closer muscle in the chela were placed ventrally in the propodite segment. A third lead is placed under the cuticle in any of the more proximal segments to serve as a ground lead. The ECG leads for the crab span the heart laterally for the best ratio in signal to noise for the recordings, and similar lead placements are made for the crayfish. The ECG leads for the crayfish and shrimp are placed in an anterior-posterior arrangement for obtaining the best signals. Modified figure from Wycoff et al. [53].

**Figure 2 animals-08-00158-f002:**
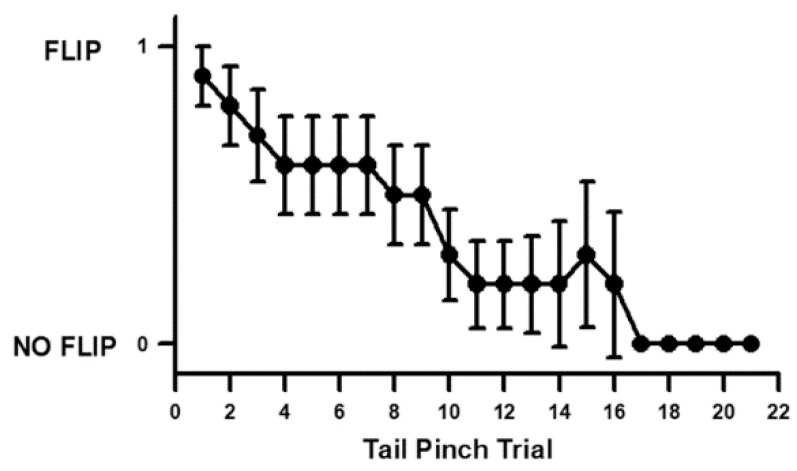
Habituation rate in tail flipping for shrimp (Belize cohort) with repetitive pinching on the telson every 30 s. On average (average +/− SEM) the animals could be pinched 16 times before habituating (30.5 °C). Ten individual shrimp were used (average +/− SEM). FLIP is a tailflip and NO FLIP is when no tail flip is observed.

**Figure 3 animals-08-00158-f003:**
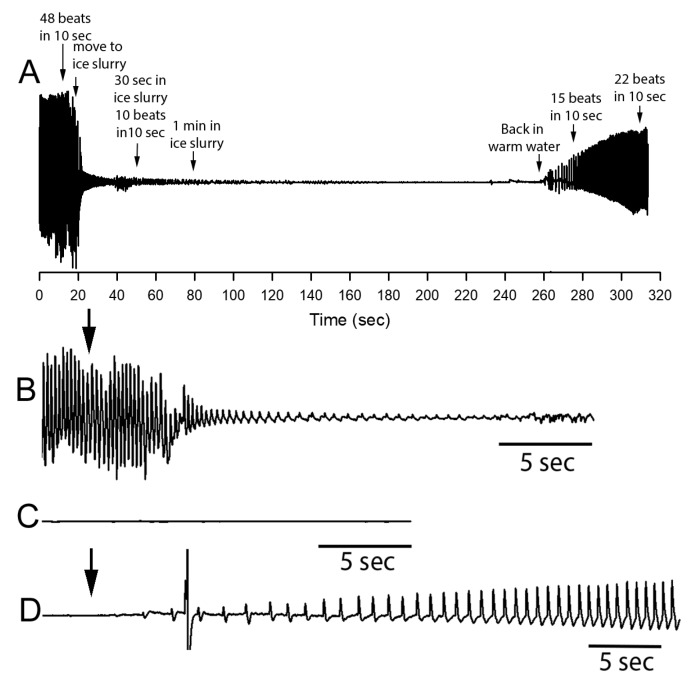
Representative electrocardiography (ECG) trace obtained from a shrimp (Belize cohort) while being immersed in a sea ice slurry (<4 °C) and during recovery. (**A**) The ECG trace before and over the short exposure to the ice slurry as well as the return to warm water is shown. (**B**) The rate of change in heart rate upon immersion in a sea ice slurry bath (arrow) is rapid. Also, note the rapid decrease in the amplitude of the signal. The shrimp was transferred from one bath (30.5 °C) to the ice slurry bath (<0.4 °C) within 2 s. The ECG trace after being in ice slurry for 1 min and 30 s is shown (**C**). Upon moving the shrimp from ice slurry back to warm water (arrow) the rate and amplitude rapidly starts to recover (**D**). All traces are shown with the same gain in signal but slightly different time scales as illustrated.

**Figure 4 animals-08-00158-f004:**
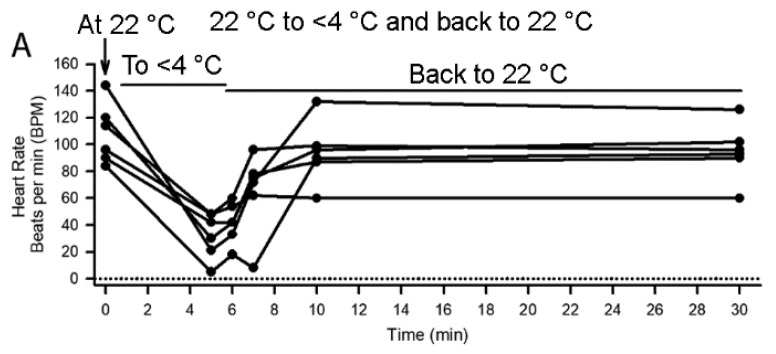
The effect of cold on heart rate for crabs, crayfish and shrimp. (**A**) Crabs decreased their heart rate upon being submerged over a 5 min period in a sea ice slurry (<4 °C). Their rates increased rapidly when placed back in warm seawater (*N* = 6). (**B**) Crayfish decreased heart rate as well upon being submerged in a fresh water ice slurry for 2 min and increased heart rate upon being returned to warm water (*N* = 6). (**C**) Five of six shrimp in the Kentucky facility decreased heart rate to no detectably beats in 30 s and within 4 min all five of the six shrimp ceased their heart rate when submerged in a sea water ice slurry (27–28 °C to <4 °C). All but one increased heart rate upon returning to warm seawater. (**D1**) The shrimp at the facility in Belize had, on average, a higher basal heart rate but also rapidly decreased heart rate upon being submerged in the sea water ice slurry (30–31 °C to <4 °C). In two of the eight the heart rate stopped in 30 s and four stopped with 2 min. The time spent submerged in the ice slurry varied from 2 to 5.5 min before returning the shrimp to warm seawater for the studies in Belize (**D2**). Note all the rates increased rapidly in warm water. Various end point measures were obtained for various animals. The rates showed a steady increase over time.

**Figure 5 animals-08-00158-f005:**
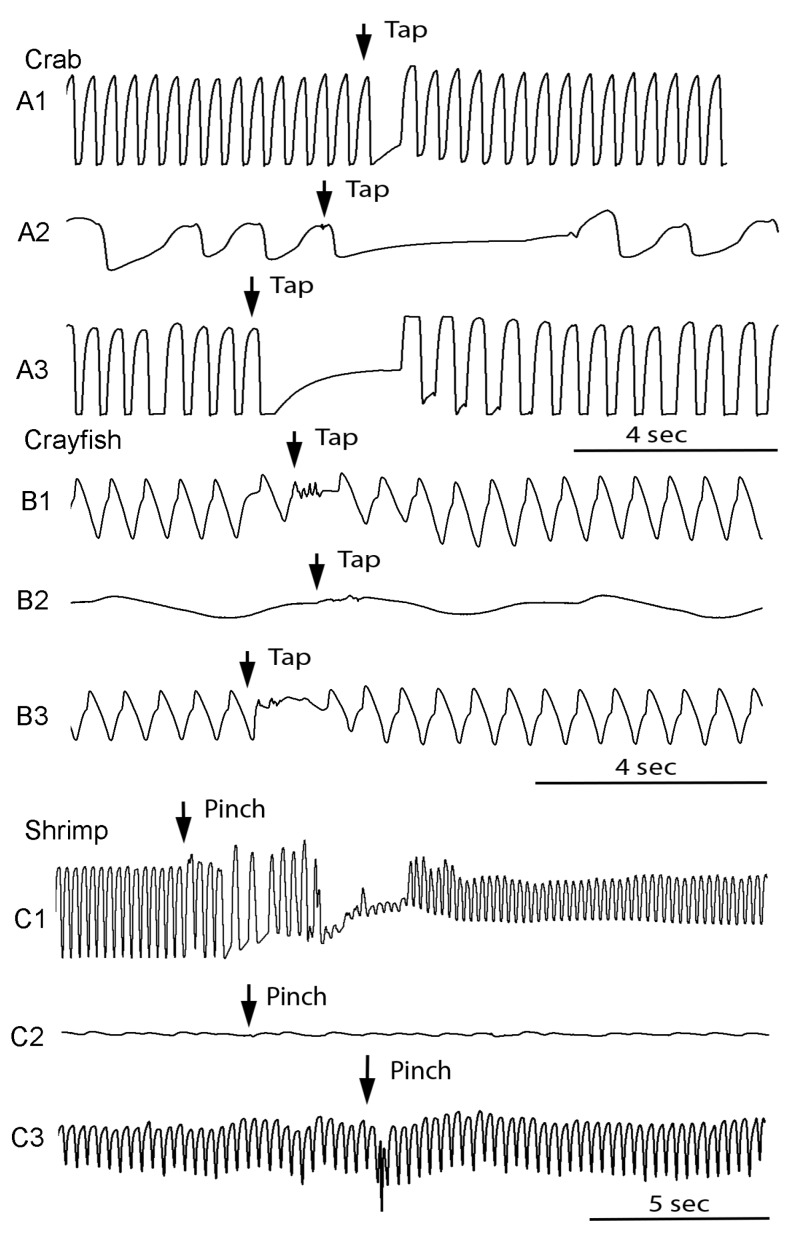
The change in heart rate with a sensory stimulus before (1), during (2), and after (3) immersion in ice slurry. Crab (**A1**), crayfish (**B1**) and Belize shrimp (**C1**) show a marked response in the ECG trace when pinched on the telson (shrimp) or tapped on the dorsal carapace (crab and crayfish) before being immersed in a sea ice slurry. When immersed in cold water no changes could be detected in the ECG with a sensory stimulus for the shrimp. Note the amplitude after ECG trace for the crayfish and shrimp is reduced after just 30 s in the cold and no observable sensory-CNS-cardiac ganglion response can be detected (**B2**,**C2**). However, for crabs there was still some responsiveness to a sensory-CNS-cardiac ganglion response even after 2 min in the ice slurry (**A2**). The heart rates of all animals rebounded quickly when placed back in the original water, and after 2 to 5 min all species showed responses to the same type of sensory stimuli (**A3**,**B3**,**C3**). The traces shown are of the same gain and from the same animal during the different paradigms.

**Figure 6 animals-08-00158-f006:**
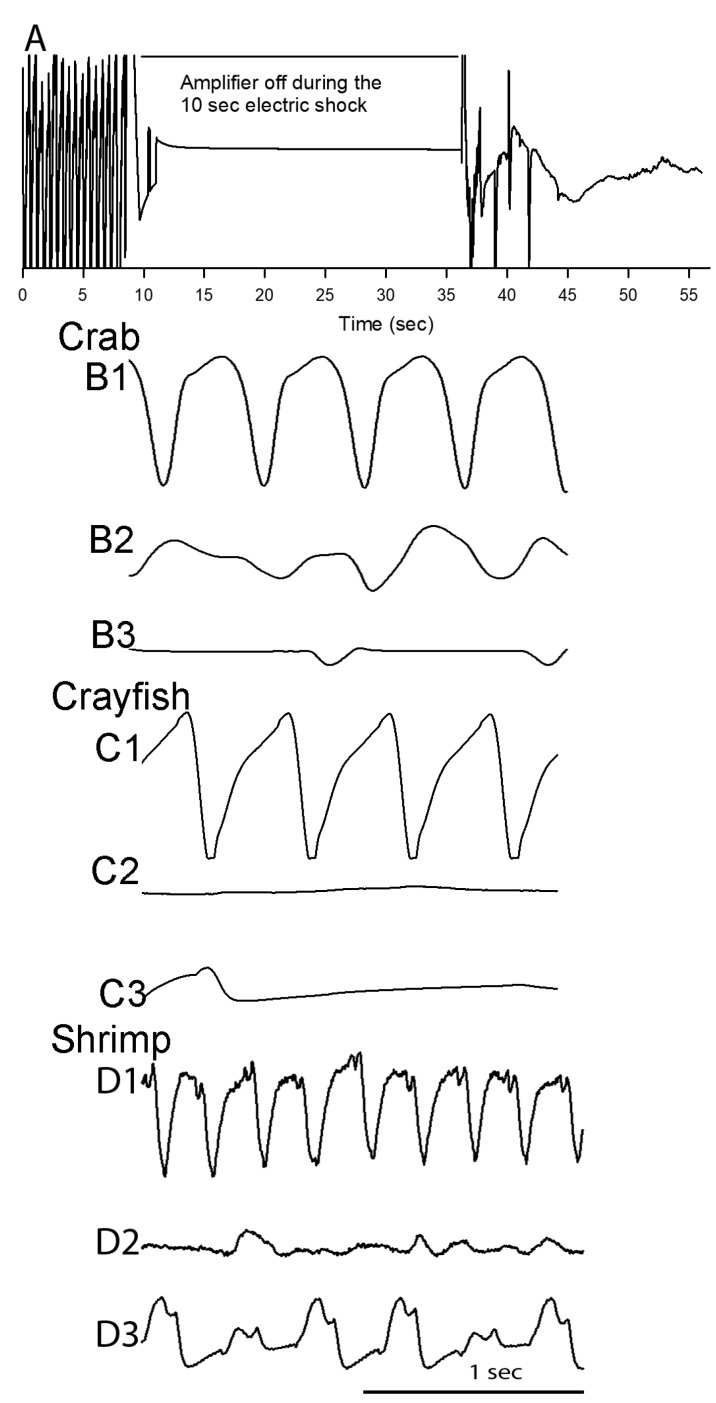
ECG traces before and immediately after electric stunning as well as after 2 to 3 min after electric stunning. (**A**) An ECG trace for a representative crayfish before and after the electric shock indicates the large change that occurs right after the shocking is over. (**B1**) Crab, (**C1**) crayfish, and (**D1**) Belize shrimp had pronounced rhythmic rates prior to electric stimulation. Arrhythmia or no measurable rate occurred immediately after electric stunning for crab (**B2**), crayfish (**C2**), and Belize shrimp (**D2**). In some cases, the arrhythmic rate persisted, and all animals showed an altered amplitude and shape in the ECG traces after electric stunning (crab, (**B3**); crayfish, (**C3**); Belize shrimp, (**D3**)). The traces shown are from the same animals during the different time periods. The 1 s scale bar applies to all traces.

**Figure 7 animals-08-00158-f007:**
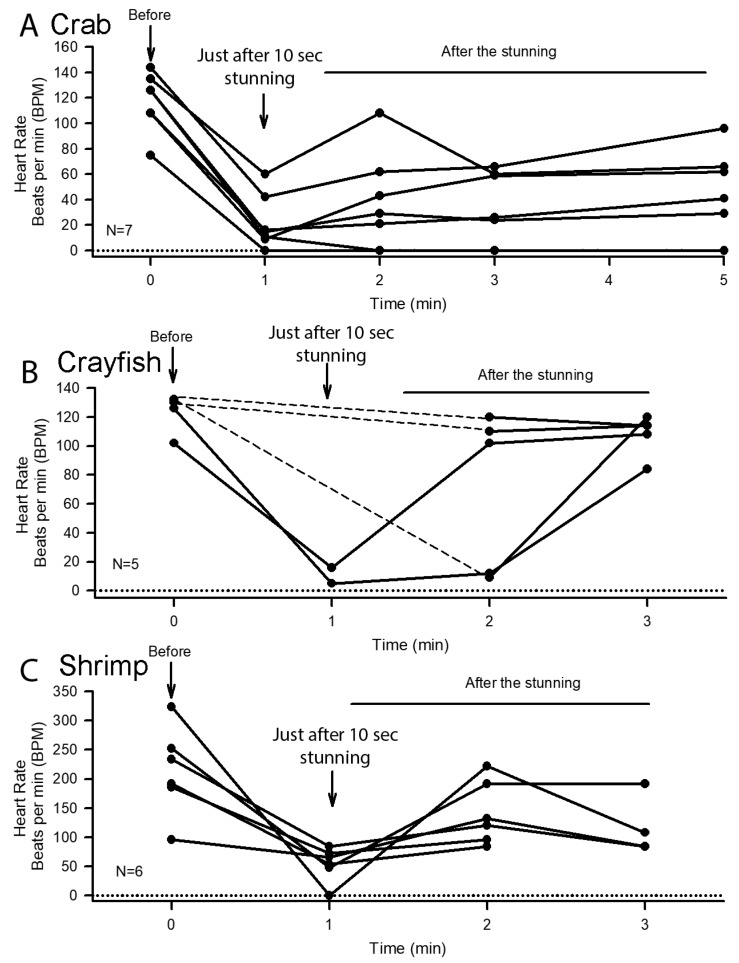
The effect of 10 s at 120 V with AC electric stunning of crab (**A**), crayfish (**B**) and shrimp (**C**). Rates were able to be obtained immediately after stunning for most animals except for some of the crayfish (dashed lines) due to electrical saturation of the amplifier during the stunning. In only one animal in each species did the heart stop beating after the 10 s window; in all but one case, for all three species, the rates increased after the electric stunning.

**Figure 8 animals-08-00158-f008:**
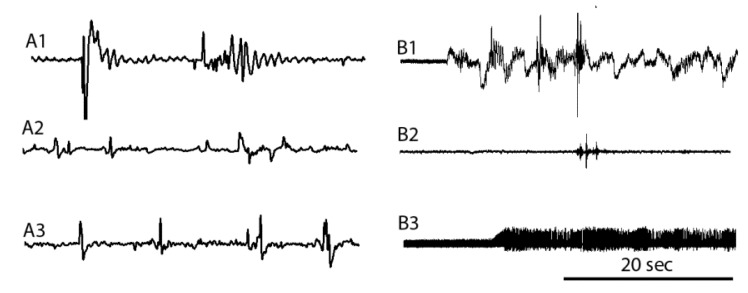
EMG traces of the closer muscle. Recordings from the chela of crab (**A)** and crayfish (**B**) before (**A1**,**B1**), during immersion in an ice slurry (**A2**,**B2**) and after (**A3**,**B3**) being returned to their original holding tanks. The closer muscle was stimulated to contract by rubbing a wooden rod on the teeth on the inside of the jaw of the chela to stimulate a sensory-CNS-motor nerve circuit. The traces shown are during the time of sensory stimulation. Note the responses are dampened while the crab (**A2**) and crayfish (**B2**) are in the ice slurry for 2 min. Upon returning the crab (**A3**) and crayfish (**B3**) to their original warmer water, the EMG traces regained their strength and maintained firing while the jaws were clamped on the wooden rod (**B3**). The traces shown are from the same crab or crayfish during the different situations. The 20 s scale bar applies to all traces.

**Figure 9 animals-08-00158-f009:**
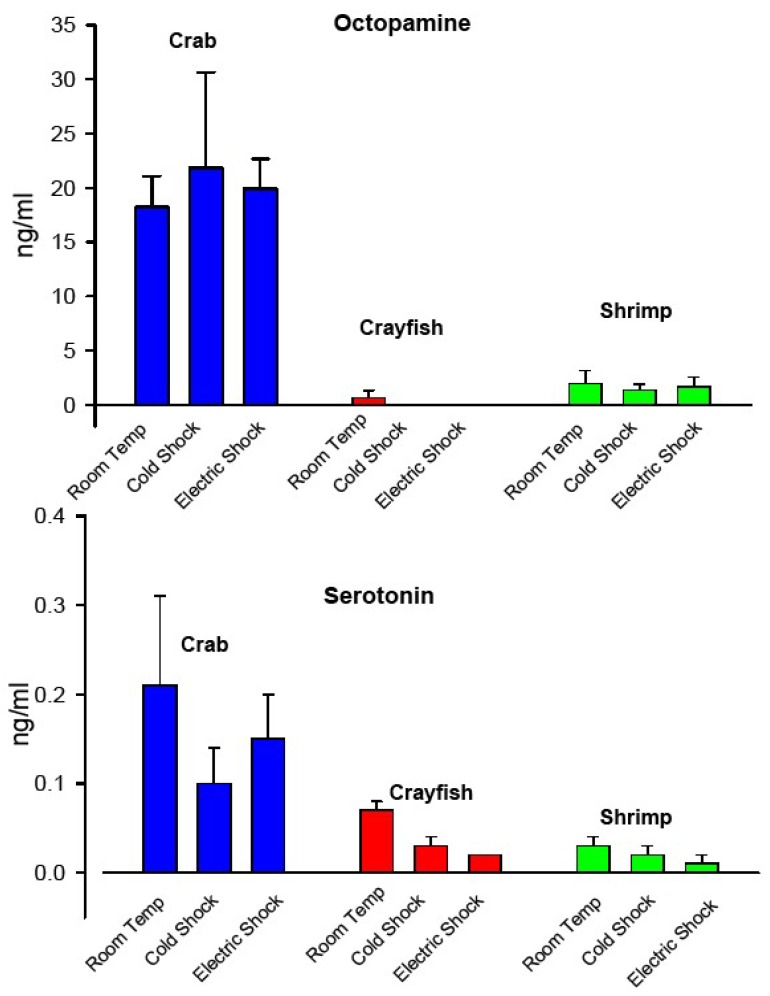
The quantification of octopamine and serotonin in the hemolymph of crab, crayfish, and shrimp at holding temperature and with immersion in ice slurry or after electric stunning. Measures were made with HPLC analysis after being immersed in an ice slurry for 5 min or after electric stunning for 10 s (120 V with AC current, 20 amps). Six animals were used for each condition. A mean (+/− SEM) value of ng/mL of hemolymph is reported for each paradigm. Statistical analysis on detectable levels for the six crayfish as compared to not being able to measure the levels with cold shock and electric stunning for crayfish produces a non-normal distribution but significant difference by Kruskal–Wallis test (*p* = 0.01). Serotonin levels were lower for cold shock (*p* = 0.028, *N* = 6, Holm–Sidak) and electric stunning (*p* = 0.004, *N* = 6, Holm–Sidak) in crayfish as compared to 20 °C but not for crabs and shrimp.

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
