# Peer review of "Physiological Changes as a Measure of Crustacean Welfare under Different Standardized Stunning Techniques: Cooling and Electroshock"

_animals, 2018, doi:10.3390/ani8090158_

Round 1

Reviewer 1 Report

This is a good paper that covers an important topic. The experiments appear to be well conducted, the results are clear and the conclusions well-founded. I only have a few queries and most of those concern the style of writing. The key problem is the use of the words "which" and "that". These are used in various ways, sometimes correctly but very often not. I will give one sentence as an example. Line 272 should have a comma before which (because the next sentence is a bit more information rather than a defining comment). Line 274 "which" should be changed to "that" because the following is the defining point. This need to be checked throughout and it will make the paper easier and more enjoyable to read. 

46 Convert pounds to kilos to keep with SI units.

68 I think you mean autotomy? If not I am not sure what you mean.

124 fresh water

170 I do not know why you try to refer to optimal heart rate. Why is it optimal? I would delete this term. This also crops up at other points in the paper.

172/3 delete "the optimal"

222 had the animals in this section been treated as in previous sections? That is had they been wired for heart rate monitoring?

315 delete "did"

316 period missing?

493 change "suboptimal" to "low"

500 change "and optimal" to "appropriate"

560 I am not sure what you mean by "the conditioning"

581 The use of the term "theory" is not appropriate in this situation. Try "idea".

585 placed?

589 reflex is not the correct term at this point. Try "activity"

590 delete in just before videos.

591 starts

603 prolonged

These are fairly minor comments. I congratulate the authors on a nice set of experiments that are very timely.

Author Response

Reviewer1:

Comments and Suggestions for Authors

1. This is a good paper that covers an important topic. The experiments appear to be well conducted, the results are clear and the conclusions well-founded. I only have a few queries and most of those concern the style of writing. The key problem is the use of the words "which" and "that". These are used in various ways, sometimes correctly but very often not. I will give one sentence as an example. Line 272 should have a comma before which (because the next sentence is a bit more information rather than a defining comment). Line 274 "which" should be changed to "that" because the following is the defining point. This need to be checked throughout and it will make the paper easier and more enjoyable to read. 

Response: Yes. Thank you for the comment and correction. We have double checked the usage of which and that throughout and noted a number of corrections.

2. 46 Convert pounds to kilos to keep with SI units.

Response: Done. Thanks

3. 68 I think you mean autotomy? If not I am not sure what you mean.

Response: Yes ….autotomy.

4. 124 fresh water

Response: Ok

5. 170 I do not know why you try to refer to optimal heart rate. Why is it optimal? I would delete this term. This also crops up at other points in the paper.

172/3 delete "the optimal"

Response: Ok. The word is removed.

6. 222 had the animals in this section been treated as in previous sections? That is had they been wired for heart rate monitoring?

Response: The animals were only used for shocking not with previous treatments such as cold exposure. This is now clarified in the revised version of manuscript.

7. 315 delete "did"

Response: Ok

8. 316 period missing?

Response: Ok

9. 493 change "suboptimal" to "low"

Response: Ok

10. 500 change "and optimal" to "appropriate"

Response: Ok

11. 560 I am not sure what you mean by "the conditioning"

Response: We see what the reviewer is refereeing to and it does seem as it was not defined well. So the text has been reworded. “These studies indicate that depending on the species of crustacean the procedures used for euthanasia or slaughter will vary and that visual observations…..”

12. 581 The use of the term "theory" is not appropriate in this situation. Try "idea".

Response: Yes, “idea” is a better word to use.

13. 585 placed?

Response: OK, the original text: “If the animals are placed in boiling water the activity also ceases rapidly…..”

14. 589 reflex is not the correct term at this point. Try "activity"

Response: OK

15. 590 delete in just before videos.

Response: OK

16. 591 starts

Response:

Orginal Line 569 “…ice slurry and it start to tail flip within a minute (S1)…”

Corrected to “…ice slurry and it starts to tail flip within a minute (S1)…

17. 603 prolonged

Response:

Orginal Line 583  “With increasing temperature and prolong heating the animals…

Corrected to “With increasing temperature and prolonged heating the animals..”

18. These are fairly minor comments. I congratulate the authors on a nice set of experiments that are very timely.

Response: We thank the review for catching many of the oversights. We changed the text accordingly. However, there were many suggestions from the other 2 reviewers which we had to address. These additional changes have altered some of the corrections above with rewording of the text.

Reviewer 2 Report

There is some good and interesting data in this paper, but they are often poorly presented. The manuscript would greatly benefit from significant revision. The paper can and should be significantly shorted (e.g., by several pages).

Line 40: This is the first of many references to “unconsciousness” or “conscious” throughout the paper. The term is never defined. “Unconsciousness” makes an unprovable inference about the internal psychological state of the animals. Words like “immobility” or “anesthesia” should be used throughout.

Line 51-54: Citations needed.

Line 54-56: This summary of issues around crustacean welfare is superficial. There are several recent references that could be included, particularly Diggles 2018.

Diggles BK. 2018. Review of some scientific issues related to crustacean welfare. ICES Journal of Marine Science: fsy058. http://dx.doi.org/10.1093/icesjms/fsy058

Sneddon LU. 2018. Comparative physiology of nociception and pain. Physiology 33(1): 63-73. https://doi.org/10.1152/physiol.00022.2017

Walters ET. 2018. Nociceptive biology of molluscs and arthropods: Evolutionary clues about functions and mechanisms potentially related to pain. Frontiers in Physiology 9(1049). https://www.frontiersin.org/article/10.3389/fphys.2018.01049

Line 62-74: Most of this could be removed and incorporated into other paragraphs.

Line 83-87: Citations needed for these definitions. The text gives examples of how stress is measured but not what “stress” is, or how one would validate a measure of stress. Support is needed that “nociception” always requires “swift reflexes”. (Indeed, the word is used only one more time in the paper, so the definition seems almost superfluous, unless a greater focus is placed on nociception – which may be a good idea). It is very important to disentangle the terms “stress,” “nociception”, and “pain”, which seem to be used interchangeably through much of the paper.

Stevens ED, Arlinghaus R, Browman HI, Cooke SJ, Cowx IG, Diggles BK, Key B, Rose JD, Sawynok W, Schwab A, Skiftesvik AB, Watson CA, Wynne CDL. 2016. Stress is not pain. Comment on Elwood and Adams (2015) ‘Electric shock causes physiological stress responses in shore crabs, consistent with prediction of pain’. Biology Letters 12(4). https://doi.org/10.1098/rsbl.2015.1006

Line 83-102: Most of this could be removed and incorporated into other paragraphs.

Line 94-96: This overlooks that the findings of Barr et al. did not replicate (Puri & Faulkes 2010), and ignores many interpretive issues (Diggles 2018).

Puri S, Faulkes Z. 2010. Do decapod crustaceans have nociceptors for extreme pH? PLoS One 5(4): e10244. https://doi.org/10.1371/journal.pone.0010244

Lines 103-114: The justification for measuring serotonin and octopamine as measures of “stress” is very weak. It appears the hypothesis is that increased levels of these two chemicals indicate stress. Diggles (2018) discusses the interpretive problems with using serotonin as a measure of stress. From reading the references provided [34-36], it appears that only Sneddon et al. (2000) measured octopamine, and the word “stress” never appears in the text of their paper. It appears these neuroactive chemicals were measured just because the authors could, not because there was any strong rationale for what they would show. All the text relating to these chemicals is so unhelpful that it could be removed entirely from the manscript.

Line 132-138: This says the point of the paper is to examine “stress.” But the paper does not clearly link the concept of stress with its dependent variables. That neural activity is measured (or its proxy, heart rate) suggest the experiments are measuring the potential for “nociception.” Since action potentials that drive heartbeats and claw reflexes are reduced, this suggests sensory neurons (such as nociceptors) would also be unable to generate action potentials. Thus, this paper actually provides relatively strong evidence that these techniques would silence nociceptive neurons and behaviours.

Line 140-195: These techniques are standard and the description of them can be shortened. It is also typical to include the species names of animals studied in the Methods.

Lines 150, 213, 214: Replace lowercase letter x with multiplication sign.

Line 170: “HR” is never defined. I presume it is an abbreviation for “heart rate.” Because “HR” has many other common meanings (e.g., “Human Resources”), this should just be spelled out in full throughout the manuscript.

Figure 1B, C: These are poor representations of a crayfish and a shrimp, particularly the abdomens.

Line 243-244: Citations needed.

Line 272-274: Omit.

Figure 2: If these are means from 10 animals, there should be error bars on the graph.

Lines 298, 316, 343, 393, 407, 439: The presentation of statistics is inadequate. The text for all reported statistics should include the sample size (only one does), the calculated test statistic with degrees of freedom, and the exact p value, not just “less than 0.5”.

Figures 3-5: It would seem to make more sense to switch Figures 4 and 5, so that Figure 3 and 4 both show shrimp data. Then, show the crab and crayfish data. For that matter, Figures 4 and 5 might be usefully consolidated, because they all show the same type of data. The lines and symbols from each individual in Figures 4 and 5 are not distinguishable in symbol or line type. Figures 4 and 5 would be more readable if they were shown as means with error bars instead of data from individuals.

Figure 6: These would be much easier for the reader to understand if these traces were presented in a 3 by 3 grid: a column each for shrimp, crab, and crayfish, with three rows for each, showing each trace.

Figure 7: Much like Figure 6, this would be improved if parts B-D were presented as a 3 by 3 grid. It is also helpful to the reader if the order of species was the same in both graphs. Figure 6 shows shrimp, crab, and crayfish, which Figure 7 shows crayfish, crab, and then shrimp.

Figure 8: Would be better presented as means with error bars.

Line 407: Normally, “trend” is used for effects that are not statistically significant.

Line 427-428, Figure 10: The text says octopamine levels for crayfish “could not be analyzed.” If this is the case, the top part of Figure 10 should not contain any crayfish data. The two halves of Figure 10 should be labelled “A” and “B”. The text described no statistical test, and the figure legend mentions a  t-test. Because t-test compare two groups, and the graphs show three (control, cold, shock), a t-test is not an appropriate statistical test for the data. I gained little from viewing these data, and they might be removed from the paper entirely.

Line 441-618: This Discussion section is far too long. Significant amounts of text could be removed from 461-558.

Line 578: It appears that rather than wrapping up, this section of the paper is describing new methods and results that were not included in the appropriate sections.

Lines 652-653, 672, 697, 700-701, 705, 709, 711, 714, 758, 771: Italicize species names.

Line 560 on: The paper misses a huge opportunity to demonstrate its relevance to actual policy. The paper shows that cooling in ice is an effective anesthetic, but using ice was specifically legislated agains in Switzerland earlier this year.

Bever L. 2018. Another country has banned boiling live lobsters. Some scientists wonder why. The Washington Post. Washington, D.C. https://www.washingtonpost.com/news/food/wp/2018/01/13/another-country-has-banned-boiling-live-lobsters-some-scientists-wonder-why/

Weintraub K. 2018. The Swiss consider the lobster. It feels pain, they decide. The New York Times. New York, New York. https://www.nytimes.com/2018/01/12/science/lobster-pain-swiss.html

https://www.blv.admin.ch/dam/blv/de/dokumente/tiere/rechts-und-vollzugsgrundlagen/faq-hummer.pdf.download.pdf/Fragen_und_Antworten_Hummer_de.pdf

https://www.blv.admin.ch/blv/de/home/tiere/tierschutz/revision-verordnungen-veterinaerbereich.html

Line 563, 584, etc: This references to “consciousness” and “cognitive function” are speculative and should be removed.

Line 578-600: The paper ignores a large (spanning about 90 years) literature on decapod crustacean tailflipping, which is particularly noticeable in this section. We know the neural circuit and what can trigger tailflips, which seems not to be considered in the description or discussion of the results.

Kramer AP, Krasne FB. 1984. Crayfish escape behavior: production of tailflips without giant fiber activity. Journal of Neurophysiology 52(2): 189-211. http://jn.physiology.org/cgi/content/abstract/52/2/189

Krasne FB, Wine JJ (1984). The production of crayfish tailflip escape responses. Neural Mechanisms of Startle Behavior. Eaton RC. New York, Plenum Press: 179-211.

Reichert H, Wine JJ. 1982. Neural mechanisms for serial order in a stereotyped behaviour sequence. Nature 296(5852): 86-87. https://doi.org/10.1038/296086a0

Wine JJ. 1984. The structural basis of an innate behavioural pattern. The Journal of Experimental Biology 112: 283-319. http://jeb.biologists.org/cgi/content/abstract/112/1/283

Edwards DH, Heitler WJ, Krasne FB. 1999. Fifty years of a command neuron: the neurobiology of escape behavior in the crayfish. Trends in Neurosciences 22(4): 153-160. https://doi.org/10.1016/S0166-2236(98)01340-X

Faulkes Z. 2008. Turning loss into opportunity: The key deletion of an escape circuit in decapod crustaceans. Brain, Behavior and Evolution 72(4): 351-361. https://doi.org/10.1159/000171488

Author Response

Reviewer 2:

1. Comments and Suggestions for Authors

There is some good and interesting data in this paper, but they are often poorly presented. The manuscript would greatly benefit from significant revision. The paper can and should be significantly shorted (e.g., by several pages).

Response: We greatly appreciate the reviewer’s insights on this topic. We reorganized the graphs and data as suggested.  The text is shortened. It was suggested to us earlier by a colleague in cutting out the ancient citations and descriptions so that is one reason for not citing the older references (90 yrs of references) as we already had 71 citations in the original manuscript. We do cite the current reviews of the field and added back our original older references to satisfy the reviewer. We are glad the reviewer suggested these additions.

2. Line 40: This is the first of many references to “unconsciousness” or “conscious” throughout the paper. The term is never defined. “Unconsciousness” makes an unprovable inference about the internal psychological state of the animals. Words like “immobility” or “anesthesia” should be used throughout.

Response: We could not agree more. We were not aware of the Diggles 2018 report prior to this submission, although we should have been. Lesson learn in updating a pubmed and Google search of a topic prior to submission.

The words “unconsciousness” or “conscious” are now replaced in the revised manuscript.

3. Line 51-54: Citations needed.

Response: This section was modified and parts deleted to make the text more concise.

 4. Line 54-56: This summary of issues around crustacean welfare is superficial. There are several recent references that could be included, particularly Diggles 2018.

Diggles BK. 2018. Review of some scientific issues related to crustacean welfare. ICES Journal of Marine Science: fsy058. http://dx.doi.org/10.1093/icesjms/fsy058

Sneddon LU. 2018. Comparative physiology of nociception and pain. Physiology 33(1): 63-73. https://doi.org/10.1152/physiol.00022.2017

Walters ET. 2018. Nociceptive biology of molluscs and arthropods: Evolutionary clues about functions and mechanisms potentially related to pain. Frontiers in Physiology 9(1049). https://www.frontiersin.org/article/10.3389/fphys.2018.01049

Response: Yes, these are excellent reviews and articles which all came out in 2018 and are now incorporated in this text.

5. Line 62-74: Most of this could be removed and incorporated into other paragraphs.

 Response: Agreed and done

6. Line 83-87: Citations needed for these definitions. The text gives examples of how stress is measured but not what “stress” is, or how one would validate a measure of stress. Support is needed that “nociception” always requires “swift reflexes”. (Indeed, the word is used only one more time in the paper, so the definition seems almost superfluous, unless a greater focus is placed on nociception – which may be a good idea). It is very important to disentangle the terms “stress,” “nociception”, and “pain”, which seem to be used interchangeably through much of the paper.

Stevens ED, Arlinghaus R, Browman HI, Cooke SJ, Cowx IG, Diggles BK, Key B, Rose JD, Sawynok W, Schwab A, Skiftesvik AB, Watson CA, Wynne CDL. 2016. Stress is not pain. Comment on Elwood and Adams (2015) ‘Electric shock causes physiological stress responses in shore crabs, consistent with prediction of pain’. Biology Letters 12(4). https://doi.org/10.1098/rsbl.2015.1006

 Response: Agreed ! Corrections to the use of the word “pain” are corrected.

7. Line 83-102: Most of this could be removed and incorporated into other paragraphs.

 Response: Yes this has been condensed and incorporated into other text

8. Line 94-96: This overlooks that the findings of Barr et al. did not replicate (Puri & Faulkes 2010), and ignores many interpretive issues (Diggles 2018).

Puri S, Faulkes Z. 2010. Do decapod crustaceans have nociceptors for extreme pH? PLoS One 5(4): e10244. https://doi.org/10.1371/journal.pone.0010244

Response: This is now addressed.

9.  Lines 103-114: The justification for measuring serotonin and octopamine as measures of “stress” is very weak. It appears the hypothesis is that increased levels of these two chemicals indicate stress. Diggles (2018) discusses the interpretive problems with using serotonin as a measure of stress. From reading the references provided [34-36], it appears that only Sneddon et al. (2000) measured octopamine, and the word “stress” never appears in the text of their paper. It appears these neuroactive chemicals were measured just because the authors could, not because there was any strong rationale for what they would show. All the text relating to these chemicals is so unhelpful that it could be removed entirely from the manuscript.

Response: Yes we understand that not a strong argument was made for measures of 5-HT and OA. And, yes we could measure them so we did as no one else measured these commonly monitored biogenic amines in the conditions we used. The data does add some to the field in understanding what physiological changes can occur or not occur with these conditions which others can add their results to in the future.   There are likely 30 or more modulators (hormones, peptides) which are known to alter physiological function in the hemolymph of crustaceans and insects. We might as well measure and report on them when one can to be able to understand what conditions can cause changes in their levels.

One argument for reporting the levels is that with longer term conditioning to cold crayfish were shown to increase octopamine levels but it is not known how rapidly this change can take place (seconds or days). So at least by reporting our current results we have shown that it appears serotonin decreases with cold and electroshock as we could measure the levels at room temperature but not in the other conditions and the average levels of OA are higher at room temperature than for cold or electroshock in the crayfish.  The data also indicates that for crabs and shrimp the changes were not as large for crayfish with environmental changes but also that the levels are quite different among the 3 species which even this by itself is interesting to report for other researchers following up on such studies.

The text has been shortened and a better justification we hope has been made for measuring the 5-HT and OA levels.

10. Line 132-138: This says the point of the paper is to examine “stress.” But the paper does not clearly link the concept of stress with its dependent variables. That neural activity is measured (or its proxy, heart rate) suggest the experiments are measuring the potential for “nociception.” Since action potentials that drive heartbeats and claw reflexes are reduced, this suggests sensory neurons (such as nociceptors) would also be unable to generate action potentials. Thus, this paper actually provides relatively strong evidence that these techniques would silence nociceptive neurons and behaviours.

Response: This is now addressed and we agree the conditions silence and reduce nociceptive neurons. This is more clearly stated in the revision.

11. Line 140-195: These techniques are standard and the description of them can be shortened. It is also typical to include the species names of animals studied in the Methods.

 Response: Agreed. As some reviewers in past, for other manuscripts, wanted every detail of already published procedures. The methods are now cited off to the published reports where they are described.

12. Lines 150, 213, 214: Replace lowercase letter x with multiplication sign.

Response: Done.

13. Line 170: “HR” is never defined. I presume it is an abbreviation for “heart rate.” Because “HR” has many other common meanings (e.g., “Human Resources”), this should just be spelled out in full throughout the manuscript.

 Response: Yikes. Ok all “HR” are now spelled out.

14. Figure 1B, C: These are poor representations of a crayfish and a shrimp, particularly the abdomens.

Response: OK. The student was so happy with her schematics. She re-did them and understands now about artistic license.

15.  Line 243-244: Citations needed.

Response: Added

16. Line 272-274: Omit.

Response: This was deleted “The rapid rate in slowing heart rate was most prevalent in shrimp. This is likely due to the relatively thin cuticle and since heart rate is starting at a high basal rate for shrimp the circulation of the hemolymph is more rapid in shrimp than in crayfish and crab.”

17. Figure 2: If these are means from 10 animals, there should be error bars on the graph.

Response: Good point. Added +/- SEM

18. Lines 298, 316, 343, 393, 407, 439: The presentation of statistics is inadequate. The text for all reported statistics should include the sample size (only one does), the calculated test statistic with degrees of freedom, and the exact p value, not just “less than 0.5”.

Response: Ok. Sure but many times I have been asked not to show the exact p value and only p<0.05, or 0.02 or 0.01 in manuscript submissions. The P values and statistics are presented as requested. 

19. Figures 3-5: It would seem to make more sense to switch Figures 4 and 5, so that Figure 3 and 4 both show shrimp data. Then, show the crab and crayfish data. For that matter, Figures 4 and 5 might be usefully consolidated, because they all show the same type of data. The lines and symbols from each individual in Figures 4 and 5 are not distinguishable in symbol or line type. Figures 4 and 5 would be more readable if they were shown as means with error bars instead of data from individuals.

Response: Well with suggestions by other reviewers and ordering the data as crab, crayfish and shrimp in the composite figures.

Good idea to combine 4 and 5 but maybe to be consistent it is good to show Crab, crayfish and shrimp data in the same order as the other figures.

The original Figure 5B2 the time points continue for 1 or 2 animals and a reader might not know what happened to the sample size with no error showing. So, we thought about this change and tried this new approach with listing the sample size next to each data point. The resultant graphs appear to be a bit misleading as the initial heart rates vary sustainably and the change per individual is lost in the averaging of the group. We even tried out graphs with percent change from initial rate but when values are very low and slight changes occur the percent change value is amplified above other animals with higher rates with smaller changes. So this becomes confusing for a reader. Also, a percent change always from the initial value or the time preceding the next data point opens more confusion for a graph over time. To resolve this issue we used a variety of symbols for each individual and enlarged the graphs. Also considering nowadays people like to see raw data so one can take values and re-graph or analyze maybe showing the individual variation can benefit someone else’s research.

20. Figure 6: These would be much easier for the reader to understand if these traces were presented in a 3 by 3 grid: a column each for shrimp, crab, and crayfish, with three rows for each, showing each trace.

Response: We liked this idea and made all new figures, the whole series, but when printing them out the figures are so compressed it is hard to see the pauses. This is one on the main points of the figures. So, we went with reviewer #3 suggestion in keeping them in order Crab, Crayfish, Shrimp and labelling the set. This way the rates can be counted by a reader as well as the pauses can be seen easily along with a good amount of the baseline rate.

21. Figure 7: Much like Figure 6, this would be improved if parts B-D were presented as a 3 by 3 grid. It is also helpful to the reader if the order of species was the same in both graphs. Figure 6 shows shrimp, crab, and crayfish, which Figure 7 shows crayfish, crab, and then shrimp.

Response: Changes were made to keep them all in order now as with the other figures.

22. Figure 8: Would be better presented as means with error bars.

Response: Well this graph has some exceptions as compared to the other line graphs related to crayfish as the recording could not be made when the amplifier was saturated. So having the line graphs allows for a clear representation of the data for each animal. Also, the line graph allows one to see the individual variation in measures which might be helpful for future researchers. We did use different symbols to help with separating out individuals.

23. Line 407: Normally, “trend” is used for effects that are not statistically significant.

Response: Yikes. Ok word use changed.

24. Line 427-428, Figure 10: The text says octopamine levels for crayfish “could not be analyzed.” If this is the case, the top part of Figure 10 should not contain any crayfish data. The two halves of Figure 10 should be labelled “A” and “B”. The text described no statistical test, and the figure legend mentions a  t-test. Because t-test compare two groups, and the graphs show three (control, cold, shock), a t-test is not an appropriate statistical test for the data. I gained little from viewing these data, and they might be removed from the paper entirely.

Response: The crayfish samples were not able to be analyzed in hemolymph for OA from crayfish during Cold shock and Electric shock due to the values being too low as compared to Room temp. This might be telling in itself as the averages also drop for 5-HT in the same conditions. In comparing. The hemolymph of crayfish could not be analyzed fully as the levels of octopamine were below the levels of HPLC detection. Since octopamine could be measured at room temperature this may indicate a reduction in their levels due to cold and electric shock. Statistical analysis on detectable levels for the 6 crayfish as compared to not being able to measure the levels with cold shock and electric stunning produces a non-normal distribution but significant difference by Kruskal-Wallis test (p=0.01). Serotonin levels were lower for cold shock (P=0.028, N=6, Holm-Sidak) and electric stunning (P=0.004, N=6 , Holm-Sidak) in crayfish as compared to 20oC but not for crabs and shrimp.

25. Line 441-618: This Discussion section is far too long. Significant amounts of text could be removed from 461-558.

Response: This text was shortened considerable in the revised text.

26. Line 578: It appears that rather than wrapping up, this section of the paper is describing new methods and results that were not included in the appropriate sections.

Response: Well this is refereeing to results in published papers to integrate to the findings we present. Anyways, this section has been reduced and a more concise closure is presented.

27. Lines 652-653, 672, 697, 700-701, 705, 709, 711, 714, 758, 771: Italicize species names.

Response: References. Ok the names are italicized now and we hope this fits the journal formatting for the references.

28. Line 560 on: The paper misses a huge opportunity to demonstrate its relevance to actual policy. The paper shows that cooling in ice is an effective anesthetic, but using ice was specifically legislated against in Switzerland earlier this year.

Bever L. 2018. Another country has banned boiling live lobsters. Some scientists wonder why. The Washington Post. Washington, D.C. https://www.washingtonpost.com/news/food/wp/2018/01/13/another-country-has-banned-boiling-live-lobsters-some-scientists-wonder-why/

Weintraub K. 2018. The Swiss consider the lobster. It feels pain, they decide. The New York Times. New York, New York. https://www.nytimes.com/2018/01/12/science/lobster-pain-swiss.html

https://www.blv.admin.ch/dam/blv/de/dokumente/tiere/rechts-und-vollzugsgrundlagen/faq-hummer.pdf.download.pdf/Fragen_und_Antworten_Hummer_de.pdf

https://www.blv.admin.ch/blv/de/home/tiere/tierschutz/revision-verordnungen-veterinaerbereich.html

Response: We were unware of this recent 2018 report by Bever. Since we did not work with lobsters we can only speculate what might be the best approach in anesthetizing them.  Fregin, T.and Bickmeyer, U. (2016) suggest slow warming might be best to anesthetize lobsters, so we reference their findings.

29. Line 563, 584, etc: This references to “consciousness” and “cognitive function” are speculative and should be removed.

Response: Yes. Agreed on and the text is adjusted throughout.

30. Line 578-600: The paper ignores a large (spanning about 90 years) literature on decapod crustacean tailflipping, which is particularly noticeable in this section. We know the neural circuit and what can trigger tailflips, which seems not to be considered in the description or discussion of the results.

Kramer AP, Krasne FB. 1984. Crayfish escape behavior: production of tailflips without giant fiber activity. Journal of Neurophysiology 52(2): 189-211. http://jn.physiology.org/cgi/content/abstract/52/2/189

Krasne FB, Wine JJ (1984). The production of crayfish tailflip escape responses. Neural Mechanisms of Startle Behavior. Eaton RC. New York, Plenum Press: 179-211.

Reichert H, Wine JJ. 1982. Neural mechanisms for serial order in a stereotyped behaviour sequence. Nature 296(5852): 86-87. https://doi.org/10.1038/296086a0

Wine JJ. 1984. The structural basis of an innate behavioural pattern. The Journal of Experimental Biology 112: 283-319. http://jeb.biologists.org/cgi/content/abstract/112/1/283

Edwards DH, Heitler WJ, Krasne FB. 1999. Fifty years of a command neuron: the neurobiology of escape behavior in the crayfish. Trends in Neurosciences 22(4): 153-160. https://doi.org/10.1016/S0166-2236(98)01340-X

Faulkes Z. 2008. Turning loss into opportunity: The key deletion of an escape circuit in decapod crustaceans. Brain, Behavior and Evolution 72(4): 351-361. https://doi.org/10.1159/000171488

Response: Agreed as we have published ourselves on the tail flip responses and teach about it in courses. We looked over these references and did not find one that mentioned cold induced the tail flip to occur. The suggested papers do describe the neural circuit but not the effect of cold in inducing a tail flip.

From personally talking to people in the crustacean processing industry, people think tail flipping is a higher neural response. So we tried to demonstrate with movies and easy to understand text that the abdomen can tail flip on its own. The audience here is not for the neurophysiologist investigating the neural circuit of tail flipping and how alpha and beta synapses vary during habituation or really how the circuit changes when the chela are removed or among dominate and submissive animals or how fast/slow 5-HT is exposed to the circuit can alter its response.

However, the reviewer’s point is made and the text cites these other references trying to make the point the circuit is well characterized and that higher processing centers are not really necessary to induce a tail flip. This is a fact which is hard for people on the floor of a processing plant of harvested crayfish and shrimp to understand when they see whole crayfish or shrimp tail flipping when they are put in an ice slurry.  We greatly appreciate the reviewer’s stance and did not mean to forget to cite original works. We appreciate any suggestions to make this a bit easier to understand for a general reader than for the expert crustacean neurobiologist whom would already know all about the circuit.

Reviewer 3 Report

This manuscript presents the results of experiments that measured heart rate and selected haemolymph parameters (octopamineand serotonin) in one shrimp, one crab and one crayfish species subjected to stunning by exposure to ice slurry or electric shock.  Although the treatments were not extensively replicated, the manuscript presents much needed empirical data upon which management decisions regarding welfare of crustaceans during handling and euthanasia, can be made.  The supplementary materials are particularly useful and informative and a worthy addition to the paper.  However, it would benefit from some editorial adjustments and tightening up of some of the terminology used to describe the behaviours and results observed.  Furthermore, in some parts of the introduction (e.g. lines 54-55, lines 93-94) and conclusions it could also benefit from a more critical assessment of previous literature on electroshocking and crustacean welfare in general in particular the claims of pain (see a recent review on this topic by Diggles (2018).  Some more specific comments include:

Terminology:  In several parts of the MS the word "consciousness" is used to describe the state of the animals response to the treatments. In animal welfare the word consciousness has other meanings and is often used to describe the theory of consciousness or sentience.  Consciousness even in humans is poorly understood.  To avoid confusing the observations here with the "hard problem" of scientifically determining the mental experiences of animals (especially invertebrates) as they respond to external stimuli (i.e. consciousness) (Browman et al. 2018).  Thus, in most contexts the authors should refrain from using the word consciousness and instead use more technically accurate descriptive terms:  i.e.

line 40 , "induce unconsciousness" should instead state "induced sedation"

line 72 , conscious = responsive

line 87, consciousness is correct term to use here in the context being discussed.

line 452, appeared unconscious = appeared stunned

line 504, assumed unconscious or dead = assumed to be dead

line 509-510,  before they regain consciousness =  irreversible stunning

line 510, induces unconsciousness = induces effective stunning

514, unconsciousness = unresponsiveness

line 562,  consciousness = stunning effectiveness

line 563, consciousness = stunning effectiveness

line 565,  unconsciousness = unresponsiveness,

line 567 "conscious of" = responsive to.

Presentation of results in figures:  While the figures are clear and useful for presenting the data,  I found it distracting that there was no uniform order for presentation of results from each species - i.e.

Figure 1 top = crab, middle = crayfish, bottom = shrimp, but

Figure 6 top = shrimp, middle = crab, bottom = crayfish,  and  

Figure 7 top = crayfish, middle = crab,  bottom = shrimp,

Figure 8  top = crab, middle = crayfish, bottom = shrimp,

I suggest that reading the results would be greatly assisted by presenting all of these figures consistently throughout as follows:  top = crab, middle = crayfish, bottom = shrimp

also, for figures 4, 5 and 8, it would assist readers if the lines/markers for individual animals were different colours or shapes so that data for individual animals can be more easily identified.

Specific comments: (strikethrough = delete, underline = add)

line 30 , "sensory perception was processed" - this is misleading as the authors did not measure the higher processing of neural signals in this experiment in  central ganglion/brain centres.  I suggest the correct wording is  (was) "to determine how neural circuits were affected during stunning....."

line 31, The circuit for sensory to the central nervous system circuit to a cardiac or skeletal muscle response was examined.

line 36,  reduced the quickest fastest in shrimp

line 38, did paralyzed them all three species.

line 39,  directly immediately after shocking in all three species,  subsequently and it increasing ed rapidly but irregularly over time but with abnormal rates

line 40 , induced unconsciousness sedation.

line 51, slaughtering crustaceans.

line 54-55, increased interest in crustacean welfare awareness and acceptance that crustaceans may be sentient beings and experience pain [4],

line 68,  automation autotomy (?)

line 72 , conscious responsive

line 86, "Pain" is an emotion that requires the capability of the animal to be aware of the noxious stimulus with the involvement of higher processing and consciousness (Rose et al. 2014).

line 89, in a humane and painless way    note: the earlier studies were speculative, as the first study confirming nociception in crustaceans was Puri and Faulkes (2015)

line 92, experience pain like vertebrates as it is difficult to provide evidence that crustaceans have this emotional capacity and awareness (Diggles 2018)

line 93-94,  However, past studies have shown the presence of opioids and their respective receptors in crabs, suggesting the existence of a pain relieving system [21].

line 97, from the antenna.  However Puri and Faulkes (2010) found no behavioural or electrophysiological evidence that antennae contained nociceptors for extreme pH or benzocaine/ethanol

line 99,  pure speculation that reduced pH of water may trigger nociceptors in crustaceans, see Puri and Faulkes 2010, Diggles 2018.  Suggest rephrase or remove.

line 113,  Zhu (2017) = [43]

line 135,  determined by a the response on of heart rate

lines 150-155.  Methods should disclose salinity and O2 concentrations for all experimental animals, not just shrimp.  Feeding frequency (once daily, ad-libitum) should also be mentioned. 

lines 207, 218  salinity/DO of seawater used to make ice slurries and hold shocked crabs/shrimp should be disclosed.

line 254,  for changes in HR in at

line 286,  sea ice slurry bath (arrow) is rapid.

line 289,  from ice slurry back to warm water (arrow)  the rate

Lines 318-325, it would be helpful if the figure caption also contained water temperature data (i.e. 27-28°C for Kentucky, 30-31°C for Belize not only the figures )

line 330,  Wilkens et al. (1974) reported central control of cardiac and scaphognathite pacemakers in a crab, please rephrase to include reference or delete.  This temporary stoppage of the heart is commonly called the startle response and is thought to have evolved to protect sediment dwelling crustaceans  from predators which can detect electric fields.

line 442-443, means of the euthanistic  euthanasia or stunning procedures in the seafood industry.

line 445-446, decrease in neural activity from the central pattern drive on cardiac ganglion driving the heart

line 452, appeared unconscious stunned

lines 458-459, this decrease is not clear and the mechanism driving it evident nor how the levels increase has not been established.

line 479, results for crabs as for the shrimp.

line 481, Chung et al. 2012 = [14]

line 492-493.  for commercial purposes [61]. This may contribute to be due to its inability to

line 495 , the water (being fresh or seawater) conducts the

line 498,  amounts of current the each animal

line 500,  and inducing paralysis effective stunning

line 503-504, at 120V to induce the make the animals motionless and to be visually assumed unconscious to be or dead.

line 509-510, for irreversible stunning before they regain consciousness.

line 510, induces unconsciousness effective stunning

line 511 or swim after electrical stunning application of the current.

line 504, time for unconsciousness unresponsiveness for a given electrical stunning setup, the handling would

line 523, heating or scaring scarring

line 534, the word blood should be replaced with hemolymph throughout (global change)

line 542, environmental stress or pain in crustaceans .  note: Stress does not equal pain, see Stevens et al. 2016

line 562,  to assess consciousness stunning effectiveness

line 563, with consciousness stunning effectiveness

line 565, render  unconsciousness unresponsiveness,

line 567 are still conscious of responsive to external

line 575, loose lose sensory

line 577, 2 min. However , fluctuations in cardiac function is depressed.

line 578, central brain ganglia

line 580, crustaceans are not to be normally

line 603, temperature and prolonged heating

line 606, and of concern for some animal welfare advocates [71, page 34]

line 611, forthcoming required to

line 615, to boiling point.  More recent studies of crayfish by Puri and Faulkes (2015) confirmed the presence of nociceptors responsive to heat,  but did not find evidence of nociceptive responses to cold water, thus bringing  the scientific rigor of some of the statements of the panel into question.  This is why recommendations should be based on rigorous scientific inquiries, rather than gross observations or assumptions (Diggles 2018).

Refs: in many locations the scientific names of the animals mentioned in the titles of the papers cited are not in italics.

References

Browman et al (2018).  Welfare of aquatic animals: where things are, where they are going, and what it means for research, aquaculture, recreational angling, and commercial fishing.  ICES Journal of Marine Science  doi:10.1093/icesjms/fsy067

Diggles BK (2018).  Review of some scientific issues related to crustacean welfare.  ICES Journal of Marine Science  doi:10.1093/icesjms/fsy058

Puri, S., and Faulkes, Z. 2010. Do decapod crustaceans have nociceptors for extreme pH? PLoS One, 5: e10244.

Puri, S., and Faulkes, Z. 2015. Can crayfish take the heat? Procambarus clarkii show nociceptive behaviour to high temperature stimuli, but not low temperature or chemical stimuli. Biology Open, 4: 441–448.

Rose et al. (2014).  2014. Can fish really feel pain? Fish and Fisheries, 15: 97–133.

Stevens et al. 2016. Stress is not pain. Comment on Elwood and Adams (2015) Electric shock causes physiological stress responses in shore crabs, consistent with prediction of pain. Biology Letters, 12: 20151006

Wilkens et al (1974).  Central control of cardiac and scaphognathite pacemakers in the crab, Cancer magister. J. Comp. Physiol. 90, 89-104.

Author Response

Reviewer 3

1. Comments and Suggestions for Authors

This manuscript presents the results of experiments that measured heart rate and selected haemolymph parameters (octopamine and serotonin) in one shrimp, one crab and one crayfish species subjected to stunning by exposure to ice slurry or electric shock.  Although the treatments were not extensively replicated, the manuscript presents much needed empirical data upon which management decisions regarding welfare of crustaceans during handling and euthanasia, can be made.  The supplementary materials are particularly useful and informative and a worthy addition to the paper.  However, it would benefit from some editorial adjustments and tightening up of some of the terminology used to describe the behaviours and results observed.  Furthermore, in some parts of the introduction (e.g. lines 54-55, lines 93-94) and conclusions it could also benefit from a more critical assessment of previous literature on electroshocking and crustacean welfare in general in particular the claims of pain (see a recent review on this topic by Diggles (2018).  Some more specific comments include:

Response: We could not agree more now since we are a bit more sensitive to the implications and concerns in this area of research. We were not aware of the Diggles 2018 report prior to this submission, although we should have been. Lesson learn in updating a pubmed and Google search of a topic prior to submission.

The words “unconsciousness” or “conscious” are now replaced in the revised manuscript.

 2. Terminology:  In several parts of the MS the word "consciousness" is used to describe the state of the animals response to the treatments. In animal welfare the word consciousness has other meanings and is often used to describe the theory of consciousness or sentience.  Consciousness even in humans is poorly understood.  To avoid confusing the observations here with the "hard problem" of scientifically determining the mental experiences of animals (especially invertebrates) as they respond to external stimuli (i.e. consciousness) (Browman et al. 2018). 

Response: The words “unconsciousness” or “conscious” are now replaced in the revised manuscript as suggested in each of the places mentioned in the text listed below.

3. Thus, in most contexts the authors should refrain from using the word consciousness and instead use more technically accurate descriptive terms:  i.e.

line 40 , "induce unconsciousness" should instead state "induced sedation"

line 72 , conscious = responsive

line 87, consciousness is correct term to use here in the context being discussed.

line 452, appeared unconscious = appeared stunned

line 504, assumed unconscious or dead = assumed to be dead

line 509-510,  before they regain consciousness =  irreversible stunning

line 510, induces unconsciousness = induces effective stunning

514, unconsciousness = unresponsiveness

line 562,  consciousness = stunning effectiveness

line 563, consciousness = stunning effectiveness

line 565,  unconsciousness = unresponsiveness,

line 567 "conscious of" = responsive to.

Response: These changes have all been made in the revised text

4. Presentation of results in figures:  While the figures are clear and useful for presenting the data,  I found it distracting that there was no uniform order for presentation of results from each species - i.e.

Figure 1 top = crab, middle = crayfish, bottom = shrimp, but

Figure 6 top = shrimp, middle = crab, bottom = crayfish,  and  

Figure 7 top = crayfish, middle = crab,  bottom = shrimp,

Figure 8  top = crab, middle = crayfish, bottom = shrimp,

Response: Agreed. The figures are now adjusted and presented better. This is a very nice suggestion and has improved the presentation. Thank you.

5. I suggest that reading the results would be greatly assisted by presenting all of these figures consistently throughout as follows:  top = crab, middle = crayfish, bottom = shrimp

also, for figures 4, 5 and 8, it would assist readers if the lines/markers for individual animals were different colours or shapes so that data for individual animals can be more easily identified.

Response: Agreed. We also are trying to adjust to the other reviewer’s concerns which dwell on the same points but with a different approach. Hopefully the revised text addresses both reviewer’s concerns on this matter.

6. Specific comments: (strikethrough = delete, underline = add)

line 30 , "sensory perception was processed" - this is misleading as the authors did not measure the higher processing of neural signals in this experiment in  central ganglion/brain centres.  I suggest the correct wording is  (was) "to determine how neural circuits were affected during stunning....."

line 31, The circuit for sensory to the central nervous system circuit to a cardiac or skeletal muscle response was examined.

line 36,  reduced the quickest fastest in shrimp

line 38, did paralyzed them all three species.

line 39,  directly immediately after shocking in all three species,  subsequently and it increasing ed rapidly but irregularly over time but with abnormal rates

line 40 , induced unconsciousness sedation.

line 51, slaughtering crustaceans.

line 54-55, increased interest in crustacean welfare awareness and acceptance that crustaceans may be sentient beings and experience pain [4],

line 68,  automation autotomy (?)

line 72 , conscious responsive

line 86, "Pain" is an emotion that requires the capability of the animal to be aware of the noxious stimulus with the involvement of higher processing and consciousness (Rose et al. 2014).

line 89, in a humane and painless way    note: the earlier studies were speculative, as the first study confirming nociception in crustaceans was Puri and Faulkes (2015)

line 92, experience pain like vertebrates as it is difficult to provide evidence that crustaceans have this emotional capacity and awareness (Diggles 2018)

line 93-94,  However, past studies have shown the presence of opioids and their respective receptors in crabs, suggesting the existence of a pain relieving system [21].

line 97, from the antenna.  However Puri and Faulkes (2010) found no behavioural or electrophysiological evidence that antennae contained nociceptors for extreme pH or benzocaine/ethanol

line 99,  pure speculation that reduced pH of water may trigger nociceptors in crustaceans, see Puri and Faulkes 2010, Diggles 2018.  Suggest rephrase or remove.

line 113,  Zhu (2017) = [43]

line 135,  determined by a the response on of heart rate

lines 150-155.  Methods should disclose salinity and O2 concentrations for all experimental animals, not just shrimp.  Feeding frequency (once daily, ad-libitum) should also be mentioned. 

lines 207, 218  salinity/DO of seawater used to make ice slurries and hold shocked crabs/shrimp should be disclosed.

Response: This was not measured as it was the same water the animals were maintained in. We could try to re-create the same conditions just for the measures if it is really necessary. The crayfish shocking was a 50:50 mix, which we state, and was just made by taking the same sea water and fresh water the animals (crab or crayfish) were housed in. Relatively straight forward to do we thought.

line 254,  for changes in HR in at

line 286,  sea ice slurry bath (arrow) is rapid.

line 289,  from ice slurry back to warm water (arrow)  the rate

Lines 318-325, it would be helpful if the figure caption also contained water temperature data (i.e. 27-28°C for Kentucky, 30-31°C for Belize not only the figures )

line 330,  Wilkens et al. (1974) reported central control of cardiac and scaphognathite pacemakers in a crab, please rephrase to include reference or delete.  This temporary stoppage of the heart is commonly called the startle response and is thought to have evolved to protect sediment dwelling crustaceans from predators which can detect electric fields.

line 442-443, means of the euthanistic  euthanasia or stunning procedures in the seafood industry.

line 445-446, decrease in neural activity from the central pattern drive on cardiac ganglion driving the heart

line 452, appeared unconscious stunned

lines 458-459, this decrease is not clear and the mechanism driving it evident nor how the levels increase has not been established.

line 479, results for crabs as for the shrimp.

line 481, Chung et al. 2012 = [14]

line 492-493.  for commercial purposes [61]. This may contribute to be due to its inability to

line 495 , the water (being fresh or seawater) conducts the

line 498,  amounts of current the each animal

line 500,  and inducing paralysis effective stunning

line 503-504, at 120V to induce the make the animals motionless and to be visually assumed unconscious to be or dead.

line 509-510, for irreversible stunning before they regain consciousness.

line 510, induces unconsciousness effective stunning

line 511 or swim after electrical stunning application of the current.

line 504, time for unconsciousness unresponsiveness for a given electrical stunning setup, the handling would

line 523, heating or scaring scarring

line 534, the word blood should be replaced with hemolymph throughout (global change)

line 542, environmental stress or pain in crustaceans .  note: Stress does not equal pain, see Stevens et al. 2016

line 562,  to assess consciousness stunning effectiveness

line 563, with consciousness stunning effectiveness

line 565, render  unconsciousness unresponsiveness,

line 567 are still conscious of responsive to external

line 575, loose lose sensory

line 577, 2 min. However , fluctuations in cardiac function is depressed.

line 578, central brain ganglia

line 580, crustaceans are not to be normally

line 603, temperature and prolonged heating

line 606, and of concern for some animal welfare advocates [71, page 34]

line 611, forthcoming required to

line 615, to boiling point.  More recent studies of crayfish by Puri and Faulkes (2015) confirmed the presence of nociceptors responsive to heat,  but did not find evidence of nociceptive responses to cold water, thus bringing  the scientific rigor of some of the statements of the panel into question.  This is why recommendations should be based on rigorous scientific inquiries, rather than gross observations or assumptions (Diggles 2018).

Response to above edits: All these changes were made but some text has also been adjusted as suggested by other reviewers. All the changes are made as fitting with the modified text. We thank the reviewer for being so thorough.

7. Refs: in many locations the scientific names of the animals mentioned in the titles of the papers cited are not in italics.

Response: References. Ok the names are italicized now and we hope this fits the journal formatting for the references.

8. References

Browman et al (2018).  Welfare of aquatic animals: where things are, where they are going, and what it means for research, aquaculture, recreational angling, and commercial fishing.  ICES Journal of Marine Science  doi:10.1093/icesjms/fsy067

Diggles BK (2018).  Review of some scientific issues related to crustacean welfare.  ICES Journal of Marine Science  doi:10.1093/icesjms/fsy058

Puri, S., and Faulkes, Z. 2010. Do decapod crustaceans have nociceptors for extreme pH? PLoS One, 5: e10244.

Puri, S., and Faulkes, Z. 2015. Can crayfish take the heat? Procambarus clarkii show nociceptive behaviour to high temperature stimuli, but not low temperature or chemical stimuli. Biology Open, 4: 441–448.

Rose et al. (2014).  2014. Can fish really feel pain? Fish and Fisheries, 15: 97–133.

Stevens et al. 2016. Stress is not pain. Comment on Elwood and Adams (2015) Electric shock causes physiological stress responses in shore crabs, consistent with prediction of pain. Biology Letters, 12: 20151006

Wilkens et al (1974).  Central control of cardiac and scaphognathite pacemakers in the crab, Cancer magister. J. Comp. Physiol. 90, 89-104.

Response: These references are now added and integrated in the modified text.

Round 2

Reviewer 2 Report

I am pleased to see some of the revisions incorporated into the manuscript. Some of the most important scientific issues, the presentation of statistics, is now resolved. I still have reservations about some aspects of the manuscript. While I appreciate the authors have expanded their explanation of the measurement of serotonin and octopamine, the revision still provides no predictions for how these chemicals should change during stress, nociception, or pain, making the results uninterpretable. I also still think some figures would benefit from revision. At a minimum, degree symbols should be added when temperature appears on a figure. The text would be improved by shortening it substantially. There are still many small typos that need correcting. Lines 107-108, 135-136, 518, 633, 639-640, 752, 781: Species names are not in italics. Lines 126, 266, 319, 450, Figure 4: Proper degree symbol needed. Line 190: Superfluous comma at end of line. Bibliography contains no reference "[6356]". Line 258: Missing period. Line 454-456: Sentence is convoluted and would benefit from revision. Line 459: "place" should be "placed". Line 608: Reference 28 to Puri and Faulkes (2010) is incorrect; should be to: Puri S, Faulkes Z. 2015. Can crayfish take the heat? Procambarus clarkii show nociceptive behaviour to high temperature stimuli, but not low temperature or chemical stimuli. Biology Open 4(4): 441-448. https://doi.org/10.1242/bio.20149654. Additionally, "cold water" should "low temperatures." Puri and Faulkes (2015) used dry ice as a cold stimulus, not cold water.

Author Response

08/04/2018

Dear Editor of Animals,

The second revised manuscript is uploaded. The review has helped make this a better manuscript.

These are the corrections made on the 2nd revised manuscript.

1. Something with the conversion from MS word to PDF did not show Italic but it was in the MS word file for each species.  This is fine if the MS word file is used.

2. This text is added for the predictions of the changes in HPLC for 5-HT and octopamine.

"

With electroshock we predicted an increase in the level of the serotonin and octopamine due to electrical stimulating the neurons to release these substances into the hemolymph. In contrast, with rapid exposure to cold we did not expect any change in the levels in the hemolymph due to decreased activity of the neurons to release these substances. In this experimental study design, we measured the concentration of

"

3. The new reference was added  #76 instead of #28

Puri S, Faulkes Z. 2015.

And this change was made :

Additionally, "cold water" should "low temperatures." Puri and Faulkes (2015) used dry ice as a cold stimulus, not cold water.

4. Degree symbols are added to text and figures on front of    "C".

5. Bibliography contains no reference "[6356]"…… in the PDF version it did not show  that 56 was a strike through (a delete)

6. Line 190: Superfluous comma at end of line…… This is now deleted.